METHODS AND RESOURCES

# Single-cell transcriptome landscape of ovarian cells during primordial follicle assembly in mice

Jun-Jie Wang[1], Wei Ge[1], Qiu-Yue Zhai[1], Jing-Cai Liu[1], Xiao-Wen Sun[1], Wen-Xiang Liu[1], Lan Li[1], Chu-Zhao Lei[2], Paul W. Dyce[3], Massimo De Felici[4], Wei Shen[1]*

1 College of Life Sciences, Institute of Reproductive Sciences, Qingdao Agricultural University, Qingdao, China, 2 Key Laboratory of Animal Genetics, Breeding and Reproduction of Shaanxi Province, College of Animal Science and Technology, Northwest A&F University, Yangling, China, 3 Department of Animal Sciences, Auburn University, Auburn, Alabama, United States of America, 4 Department of Biomedicine and Prevention, University of Rome Tor Vergata, Rome, Italy

* wshen@qau.edu.cn, shenwei427@163.com

**Data Availability Statement:** The accession number of ovarian single-cell RNA sequencing raw data reported in this paper is NCBI GEO: GSE134339. The R code using for data processing

## Abstract

Primordial follicle assembly in the mouse occurs during perinatal ages and largely determines the ovarian reserve that will be available to support the reproductive life span. The development of primordial follicles is controlled by a complex network of interactions between oocytes and ovarian somatic cells that remain poorly understood. In the present research, using single-cell RNA sequencing performed over a time series on murine ovaries, coupled with several bioinformatics analyses, the complete dynamic genetic programs of germ and granulosa cells from E16.5 to postnatal day (PD) 3 were reported. Along with confirming the previously reported expression of genes by germ cells and granulosa cells, our analyses identified 5 distinct cell clusters associated with germ cells and 6 with granulosa cells. Consequently, several new genes expressed at significant levels at each investigated stage were assigned. By building single-cell pseudotemporal trajectories, 3 states and 1 branch point of fate transition for the germ cells were revealed, as well as for the granulosa cells. Moreover, Gene Ontology (GO) term enrichment enabled identification of the biological process most represented in germ cells and granulosa cells or common to both cell types at each specific stage, and the interactions of germ cells and granulosa cells basing on known and novel pathway were presented. Finally, by using single-cell regulatory network inference and clustering (SCENIC) algorithm, we were able to establish a network of regulons that can be postulated as likely candidates for sustaining germ cell-specific transcription programs throughout the period of investigation. Above all, this study provides the whole transcriptome landscape of ovarian cells and unearths new insights during primordial follicle assembly in mice.

is publicly available at GitHub at https://github.com/WangLab401/2020scRNA_murine_ovaries.

**Funding:** This work was supported by National Key Research and Development Program of China (2018YFC1003400), National Nature Science Foundation (31671554 and 31970788) and Taishan Scholar Construction Foundation of Shandong Province (ts20190946). The funders had no role in study design, data collection and analysis, decision to publish, or preparation of the manuscript.

**Competing interests:** The authors have declared that no competing interests exist.

**Abbreviations:** AHR, aryl hydrocarbon receptor; BMPs, bone morphogenetic protein; BPGs, bipotential pre-granulosa cells; D-BPG, differentiating bipotential pre-granulosa cell; D-EPG, differentiating epithelial pre-granulosa cell; E, embryonic day; EIF4A1, eukaryotic translation initiation factor 4A1; EPGs, epithelial pre-granulosa cells; FIGLA, folliculogenesis specific basic helix-loop-helix transcription factor; FOXL2, forkhead box L2; FOXO3, Forkhead box O3; FSH, follicle-stimulating hormone; G3BP2, GTPase-activating protein-binding protein 2; GDF9, growth differentiation factor 9; GO, Gene Ontology; GRN, global research network; HLH, helix-loop-helix; HSPB11, heat shock protein family B member 11; IRX, iroquois homeobox; KEGG, Kyoto Encyclopedia of Genes and Genomes; KITL, KIT proto-oncogene, receptor tyrosine kinase ligand; LATS, large tumor suppressor kinase; LGR5, leucine-rich repeat-containing G-protein coupled receptor 5; LHX8, LIM homeobox 8; MVH, mouse vasa homologue; NOBOX, NOBOX oogenesis homeobox; OOEP, oocyte expressed protein; OR, ovarian reserve; PC, principal component; PD, postnatal day; PF, primordial follicle; PGC, primordial germ cell; PI, propidium iodide; POF, premature ovarian failure; PVDF, polyvinylidene fluoride; RA, retinoic acid; RSPO1, R-spondin-1; SCENIC, single-cell regulatory network inference and clustering; SCMC, subcortical maternal complex; scRNA-seq, single-cell RNA sequencing; SMAD, homologues of *Drosophila* protein, mothers against decapentaplegic and *Caenorhabditis elegans* protein sma; SOHLH, spermatogenesis and oogenesis specific basic helix-loop-helix; TAF4b, TATA-box binding protein associated factor 4b; TAZ, tafazzin family protein; TF, transcriptional factor; TGF-beta, transforming growth factor-beta; t-SNE, t-distributed stochastic neighbor embedding; UMAP, uniform manifold approximation and projection; WNT4, Wnt family member 4; Yap1, yes-associated protein 1.

# Introduction

Gametogenesis is a finely regulated and complex process beginning from germline specification that gives rise to primordial germ cells (PGCs) and ending with either mature oocytes or sperms. In female mammals, primordial follicles (PFs) are the first functional unit of reproduction, each comprising a single oocyte surrounded by supporting somatic cells termed granulosa cells. Within 1 week after birth, the PFs constitute the ovarian reserve (OR), responsible for continued folliculogenesis and oocyte maturation throughout the adult life [1,2]. The PF population has long been believed to be nonrenewable, although this notion has recently been challenged [3,4]. Putting the latter discussion aside, it is evident that during the reproductive life span, the number of PFs progressively diminishes due to atresia as well as recruitment, maturation, and ovulation. In primates, the depletion of the reserve is the major determining driver of menopause in adult females and of various conditions of infertility in younger individuals grouped as having premature ovarian failure (POF) [5].

In the mouse, germline specification occurs before gastrulation in the epiblast at around embryonic day (E) 6.25, and this is primarily in response to bone morphogenetic protein (BMP) signals [6,7]. Around E7.0 to E8.0, PGCs are identifiable in the extraembryonic mesoderm at the angle between the allantois and the yolk sac [8]. From here, PGCs begin to migrate, and most of them move into the embryo proper to initiate colonization by E10.5, between E11.5 and E12.5, and the gonadal ridges form from the coelomic epithelium [9,10]. After active proliferation, PGCs/oogonia cease dividing at E13.5 and enter meiosis with the resulting formation of oocytes [11]. The oocytes are closely associated in clusters, termed germ cell nests, in which they are connected through cytoplasmic bridges formed because of incomplete cytokinesis during mitosis [12]. Oocyte loss and nest breakdown begin after birth in the cortical region of the ovary, but in the medullar region, these processes begin as early as E17.5 [13]. Nest breakdown begins after oocytes undergo the first meiotic block at the diplotene stage of prophase I. The failure of mouse oocytes to reach the diplotene stage impairs this process and consequently PF assembly [14,15]; however, it is still understood the 2 events of meiotic progression and PF formation were independent [16]. As reported above, germline nest breakdown is associated with a massive loss of oocytes that, in the mouse, has been reported to occur following programmed cell death mainly in the form of apoptosis and autophagy [15,17,18]. The surviving single oocytes become enveloped by granulosa cells from the surrounding nest, thereby producing the PFs. Synchronized development of oocytes and granulosa cells, as well as their interactive communication are required for efficient nest breakdown and the subsequent PF assembly.

Both in mouse and human, mutations of *Wnt4* and *Rspo1* genes induce failure of pre-granulosa cell differentiation, impair nest breakdown, and finally result in POF [19]. Recent studies in the mouse identified 2 waves of granulosa cell differentiation that contribute to 2 discrete populations of follicles [20,21]. The precursors of both populations arise from the ovarian surface epithelium. The first population, identifiable by the early expression of the transcription factor FOXL2, encircles the germ cell nests and eventually form the earliest PFs that develop within the ovarian medulla and undergo rapid maturation and atresia. The second population is composed of cells expressing leucine-rich repeat-containing G-protein coupled receptor 5 (LGR5), a likely receptor of Wnt family member 4 (WNT4)/R-spondin 1 (RSPO1), that subsequently express forkhead box L2 (FOXL2) and down-regulate LGR5 and will form the OR of PFs in the cortex [22,23]. Other somatic cell transcription factors, such as iroquois homeobox (IRX)-3 and IRX-5, and the folliculogenesis specific basic helix-loop-helix transcription factor (FIGLA), spermatogenesis and oogenesis specific basic helix-loop-helix (SOHLH)-1, and SOHLH-2, LIM homeobox 8 (LHX8), NOBOX oogenesis homeobox

(NOBOX), TATA-box binding protein associated factor 4b (TAF4b), and aryl hydrocarbon receptor (AHR) also appear to play a role in nest breakdown and PF assembly both in mice and humans [24,25]. Several signaling exchanges between oocytes and pre-granulosa cells are also implicated in such processes, such as Notch, KIT proto-oncogene, receptor tyrosine kinase ligand (KITL)/KIT system, neurotrophins, transforming growth factor-beta (TGF-beta) and its family members growth differentiation factor 9 (GDF9), BMP15, Activin A, inhibin, follistatin, and anti-Mullerian hormone, as well estrogens, progesterone, and finally follicle-stimulating hormone (FSH) [1,15].

Despite these results and previous transcriptome analyses performed in rat [26,27] and mouse fetal and early postnatal ovaries [28,29], the precise molecular mechanisms underlying oocyte survival/death, granulosa cell differentiation, and the crosstalk among them during PF assembly are still incomplete. Moreover, all the latter studies were performed on the entire ovary or follicles so that the contribution of each cell type to the transcriptomes was not distinguishable. The recent advent of single-cell RNA sequencing (scRNA-seq) technologies makes it possible to identify specific cell subpopulations and the genetic programs regulating their differentiation in reproductive organs [10,30].

In the present paper, to achieve the mentioned goals, we focused on PF assembly in the mammalian ovary by performing scRNA-seq analyses on late fetal and early postnatal murine ovaries. The results and bioinformatics information reported here will certainly be useful for elucidating the molecular mechanisms underlying PF assembly and for selecting candidate regulatory factors for further investigation.

## Results

### scRNA-seq identified several types and subpopulations of germ cells and somatic cells in late fetal and early postnatal ovaries

To trace the distinct cell lineages and characterize the gene expression dynamics of ovarian cells during the crucial period of ovarian development lasting from late fetal and early postnatal age, ovaries were collected from E16.5, postnatal day 0 (PD0) and 3 (PD3) mice and subjected to immunohistochemical staining and scRNA-seq analyses. As expected, results from stained sections showed that the germ cell nests gradually broke down, while the percentage of germ cells enclosed in PFs increased from 9.95 ± 1.38% at E16.5 to 46.02 ± 1.46% at PD3 (S1A and S1B Fig).

Ovarian cell populations were then prepared and barcoded following single cell capture, and RNA sequencing was performed (Fig 1A). Followed filtering out of low-quality cells (including empty droplets, dying cells, and potential doublets or multiplets), the remained number of cells were 5,742 in E16.5; 5,255 in PD0; and 5,871 in PD3, and the mean number of genes expressed in each sample was 2,258, 2,844 and 2,768, respectively (S1C Fig). According to uniform manifold approximation and projection (UMAP), the cell populations at 3 developmental stages were clustered (Fig 1B), and 7 cell types were identified, and according to their distinct transcriptomic signatures (S1 Table), they could be subdivided as follows: germ cells, granulosa cells, stromal cells, erythrocytes, immune cells, endothelial cells and epithelial cells (Fig 1C and 1D and S1D Fig). The expression of representative genes known to be specific for each cell type was reported in Fig 1E. In S1E Fig, the percent changes within the ovarian populations of the 7 identified cell types, during the analyzed developmental period, were reported. Following, the differentiated fate was examined mainly focusing on 2 major participants for PF formation, germ cells, and granulosa cells.

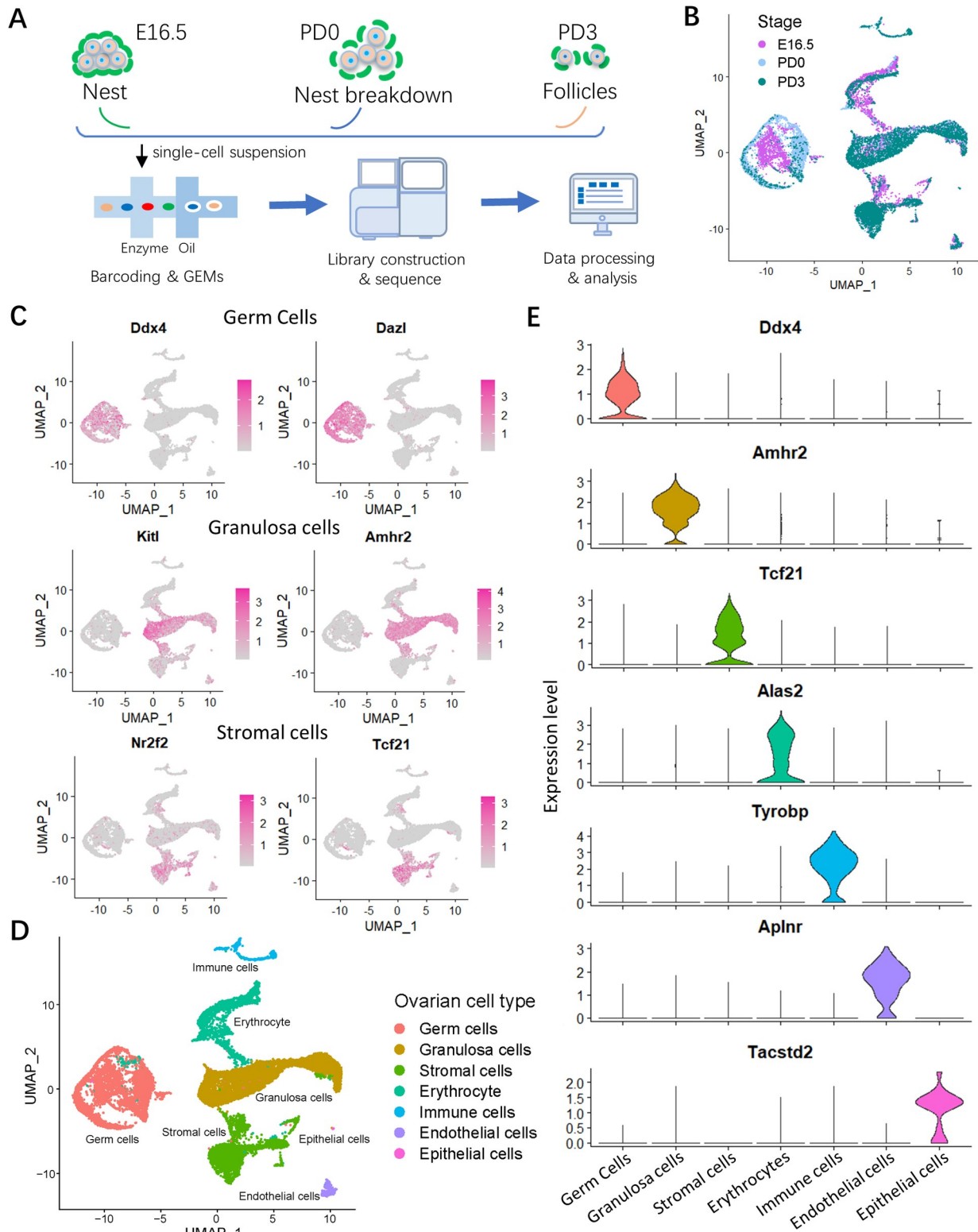

**Fig 1. scRNA-seq identified ovarian cell types during PF formation.** (**A**) Schematic diagram of the scRNA-seq analysis procedure. Ovaries were isolated and disaggregated into single cell suspensions; cells were barcoded and used for library construction onto 10× Genomics platform and the data produced after sequencing analyzed by dedicated software. (**B**) UMAP plot of ovarian cells colored by the selected developmental time points (that is Stage). E16.5 for medium orchid, PD0 for light blue and PD3 for dark cyan. (**C**) Feature plots of specific marker genes of

different cell types, including germ cells, granulosa cells, and stromal cells. The detailed marker genes of each cell type are shown in S1 Table. (**D**) UMAP plot of 7 ovarian cell types. (**E**) Vlnplots of specific marker gene in each ovarian cell type. The sequencing data was deposited availably in GSE134339, and this figure can be produced using scripts at https://github.com/WangLab401/2020scRNA_murine_ovaries. scRNA-seq, single cell RNA-sequencing; UMAP, uniform manifold approximation projection; E16.5, embryonic day 16.5; PD0, postnatal day 0; PD3, postnatal day 3.

## Genetic dynamics of female germ cells from the nest stage to the primordial follicle formation

To dissect the heterogeneity of germ cells during development in the period of investigation through UMAP analysis, the germ cell populations were identified as 5 clusters: cluster 2 cells were mostly present at E16.5; clusters 1, 3, and 4 were mostly present at PD0; and cluster 5, at PD3 (Fig 2A). To obtain dynamic pattern of gene signature, the germ cell cluster-specific marker genes were identified (S2 Table). The top 10 expressed genes of each cluster were shown in a heatmap in Fig 2B. According to these analyses, a genetic dynamic model, including pre-, early- and late-follicle formation stages during germ cell development was drawn (Fig 2C and 2D). Genes known to be expressed throughout female germ cell development at the beginning of meiosis are *Stra8*, *Prdm9*, and *Meioc* [31–34]; during PF formation are *Figla*, *Lhx8*, *Sohlh1*, *Nobox*, and *Ybx2* [4, 15, 24]; and for early oocyte growth are *Gdf9*, *Zp2*, and *Zp3* [35–38] (Fig 2B and 2D). Importantly, several new genes coupled with development stages of germ cells at the pre- (i.e., *M1ap*, *Hspb11*, and *Pigp*), early (i.e., *Eif4a1*, *G3bp2*, *Id1*, *Acat1*, and *Ldhb*), and late (i.e., *Ooep*, *Gm15389*, and *Padi6*) follicle formation stages were identified (Fig 2D). Moreover, these genes are also consistent with developmental time points (S2A Fig). In addition, the percentage changes of cells at these stages (pre-, early- and late-follicle formations) from E16.5 to PD3 were shown in S2B Fig.

To improve the reliability of the dataset, the expressions of heat shock protein family B member 11 (HSBP11), GTPase-activating protein-binding protein 2 (G3BP2), and oocyte expressed protein (OOEP) representing the markers of pre-, early- and late-follicle formation stages, respectively, were confirmed by immunohistochemistry (Fig 3A–3C). Importantly, the same expression trend of these proteins was also detected with western blot, as well as eukaryotic translation initiation factor 4A1 (EIF4A1) for early follicle formation stage (Fig 3D).

## Fate transition of oocytes along pseudotime trajectories

To dissect the fate determination of germ cells throughout the investigated period, they were ordered along pseudotime trajectories according to the gene expression reported above. Three states (a jargon in pseudotime analysis; "state" is assigned to mark the segment of the trajectory tree in Monocle) were obtained, in which the majority of cells at the pre-follicle formation stage belonged to state 1, termed as "germ cells in nest", while those at the early and late follicle formation stages were in states 2 and 3; in particular, cells of state 3 were "germ cells within follicle" (Fig 4A–4C). Meanwhile, the expressions of representative genes (marker genes of 3 follicle formation stages identified above) along with pseudotime trajectories were consistent with our inference (S2C Fig). In such pseudotime trajectories, 3 branches implied 3 differentiated stages of germ cells; moreover, 5 gene sets with distinct patterns were identified (S3 Table), which were likely involved in the commitment of germ cells within nests into follicles; clusters 1, 2, 3, 4, and 5 including 722; 1,819; 359; 1,077; and 1,904 genes, respectively (Fig 4D and 4E). The heatmap showed the dynamics of gene expression in the germ cells at the nest and follicle stages: Set 1 includes genes with steady high expression in follicular oocytes; set 2 genes showed an anabatic high expression in follicular oocytes; these 2 gene sets were both slightly or scarcely expressed at germ cells in nests (or state 1); set 3 seemed to function mainly at pre-

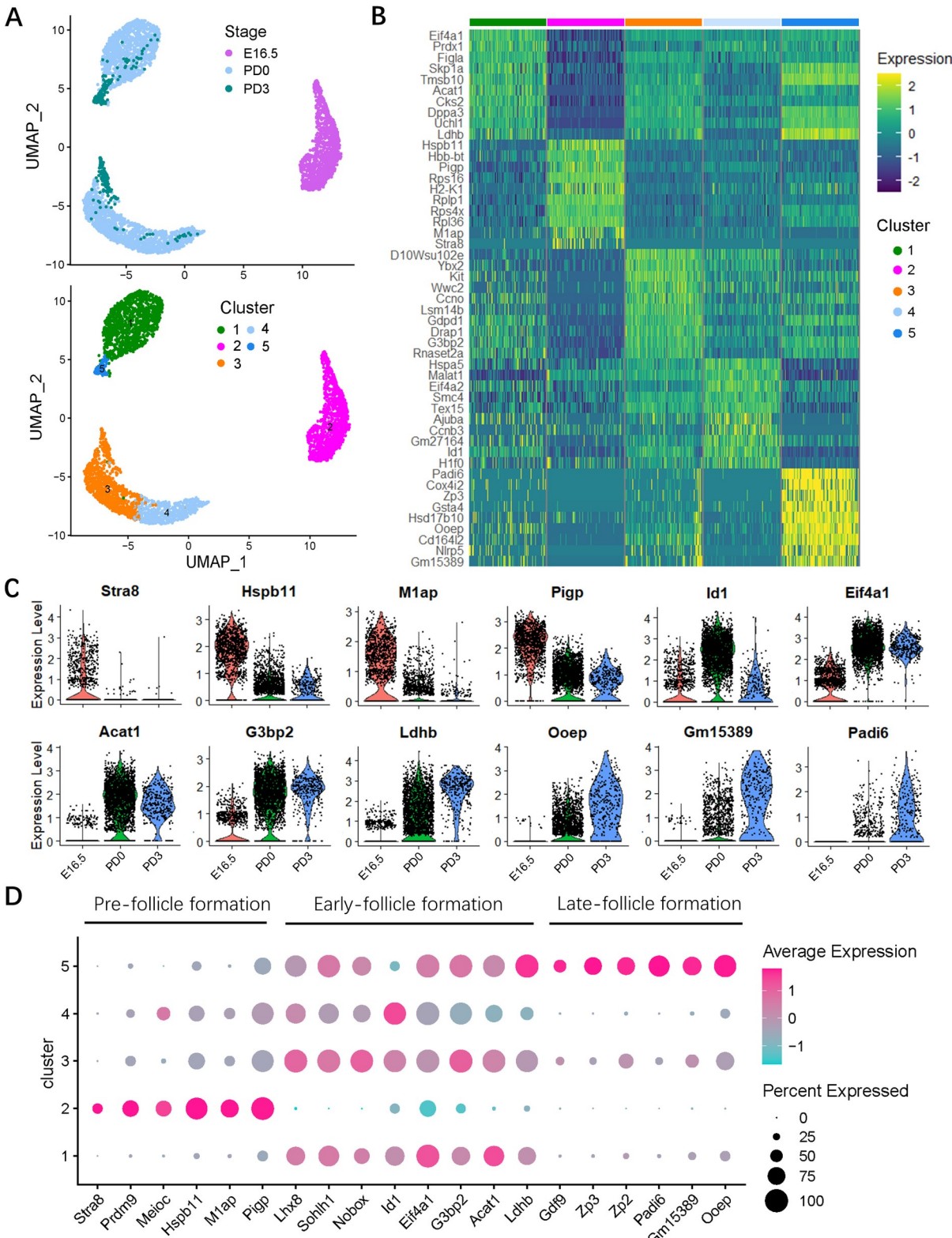

**Fig 2. Molecular characterization of the germ cell subsets.** (**A**) Cluster analysis of germ cells with UMAP plots based on developmental timeline (upper) and transcriptional patterns (below). (**B**) Heatmap of top 10 marker genes of the germ cell clusters. Top 50 marker genes in each cluster are shown in S2 Table. (C) Vlnplots of the expression level of representative genes along with developmental timeline. (D) Dot plot of general and stage-specific (pre-, early- and late-follicle formations) germ cell marker genes expressed in cell clusters. Dot size represents the

cell percentage expressed in each cluster; dot color marks the average expression level of each gene. The sequencing data was deposited availably in GSE134339, and this figure can be produced using scripts at https://github.com/WangLab401/2020scRNA_murine_ovaries. UMAP, uniform manifold approximation projection.

and early follicle formation stages and was significantly decreased for germ cells within follicles; sets 4 and 5 comprised genes specifically up-regulated in germ cells in nests. The expression kinetics of each gene set are shown in Fig 4E. Enriched Gene Ontology (GO) terms of the most expressed 100 genes of these gene sets are shown in Fig 4F (S4 Table). For example, gene set 1 enriched "circadian regulation of gene expression, piRNA metabolic process, female gonad development, and cell division"; gene set 2 contained genes involved in "histone deacetylation, and cell cycle phase transition"; gene set 3 enriched genes in "chromatin organization, mRNA processing, sister chromatid cohesion, and cellular response to DNA damage stimulus"; and gene set 4 enriched genes of "regulation of chromosome organization, gene silencing, mRNA metabolic process, and in utero embryonic development", as well as "chromatin organization"; importantly, gene set 5 were most related to "female gamete generation, DNA repair, regulation of RNA splicing, and regulation of gene expression, epigenetic." Meanwhile, the representative genes in each set are displayed with pseudotime trajectories in Fig 4G. Specifically, representative genes enriched in GO terms of "histone deacetylation" (*Hdac2* and *Morf4l1*) and "regulation of gene expression and epigenetic" (*Morc2a* and *Smarca4*) were highlighted (S2D and S2E Fig). To further characterize the germ cell fate transition from nests to follicles, gene sets 1 and 2 that were dramatically increased in follicular germ cells were merged and analyzed with Kyoto Encyclopedia of Genes and Genomes (KEGG) pathway, and the results showed abundant transcripts for "RNA transport, spliceosome, oxidative phosphorylation, nucleotide excision repair, and citrate cycle (TCA cycle)" (S2F Fig). Furthermore, KEGG analysis of genes in sets 4 and 5 that were actively transcribed in germ cells of the nest stage enriched "ribosome, regulation of actin cytoskeleton, focal adhesion, and adherens junction," and several pathways, including "Ras, Rap1, mTOR, Wnt, insulin, and AMPK signaling" (S2G Fig and S5 Table).

## Mapping oocyte-specific regulon networks by SCENIC

We further investigated the regulons (i.e., transcriptional factors (TFs) and their target genes) activity of germ cell-specific TFs using SCENIC, an algorithm developed to deduce global research networks (GRNs) and the cellular status for scRNA data [39]. Based on 201 regulons activity with 9,208 filtered genes with default filter parameters, the selected cells were clustered with cell state (right, pseudotime analysis label) and developmental stages (left) in t-distributed stochastic neighbor embedding (t-SNE) method calculated by SCENIC (S3A Fig), and the regulons density was also mapped to t-SNE plots (S3B Fig). Accordingly, regulon activity was binarized and matched with germ cell state and developmental stages (left panel); in addition, some representative regulons and their motifs (right panel) were listed (Fig 5A). As shown, a series of TFs displayed a more dynamic pattern. For example, *Nr3c1*, *Brca1*, *Kdm5a*, and *Kdm5b* were active mainly at the germ cell nest stage (Fig 5B and S3C Fig), and *Klf7* and *Gabpa* seemed to be mostly activated at the nest stage, but gradually turned off as follicular assembly occurred (Fig 5C). Similarly, *Smarcb1*, *Hes1*, *Sox15*, and *Stat3* were mainly associated with states 2 and 3, corresponding to the period of PF assembly (Fig 5D and S3D Fig). Furthermore, a portion of TFs appeared to serve as a transient switch, such as *Foxo1* had a limited role in germ cells within nests and *Foxo3* functioned during a period of the follicular stage (Fig 5E). Moreover, several TFs, such as *Jun* and *Klf2*, were significantly represented during all oocyte stages, which supports a role for these TFs as likely candidates for sustaining a germ cell-specific transcription program throughout the investigated period (S3E Fig).

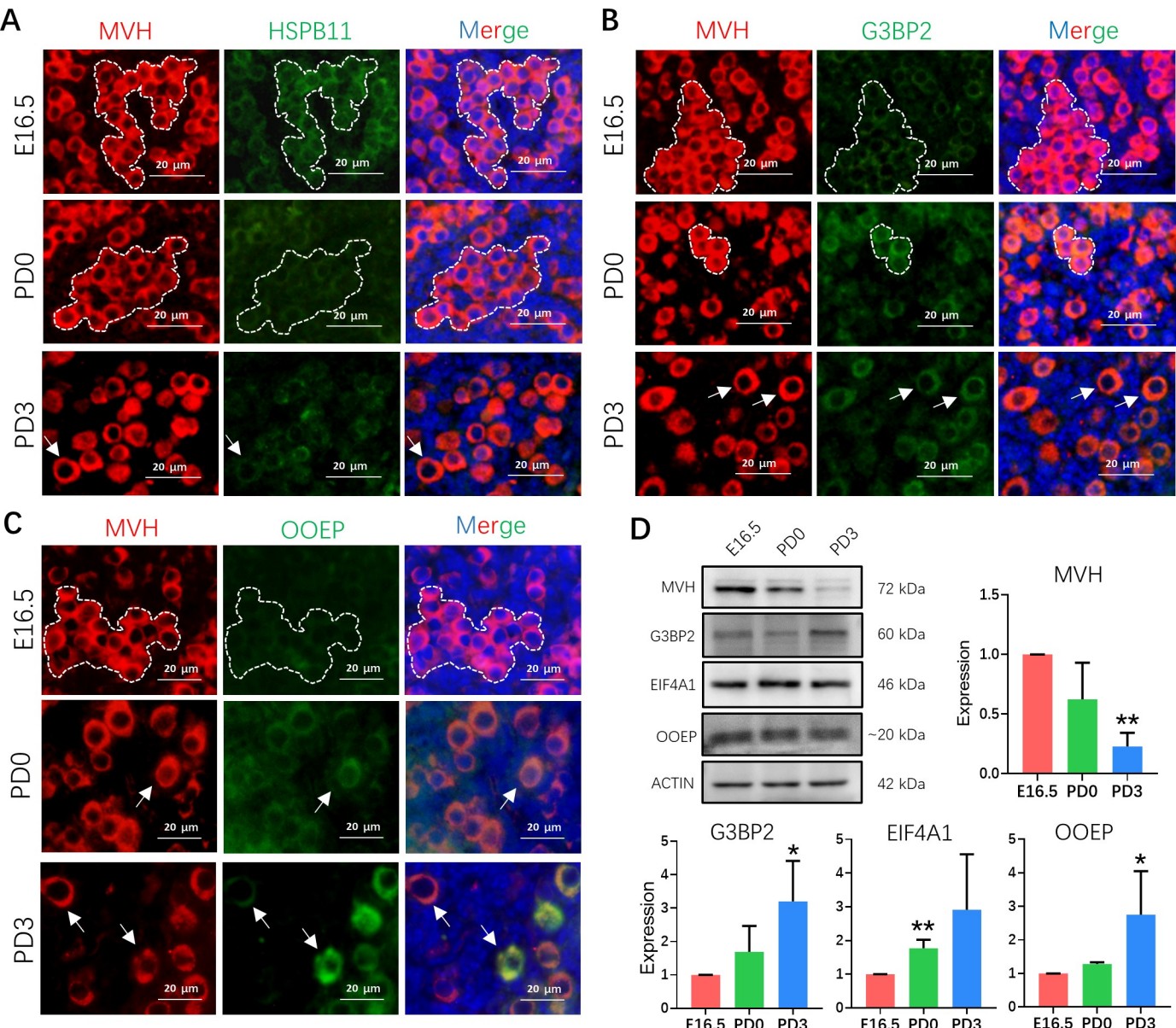

**Fig 3. Marker gene detection of germ cells at pre-, early- and late-follicle formation stages.** (**A**) Representative images of the HSPB11 staining of ovarian sections from E16.5 embryos, and PD0 and PD3 pups. Germ cells were marked by MVH (red); HSPB11 was stained with green; and nucleus was counterstained with Hoechst 33342 (blue). Dotted white line label the germline nest and white arrow mark the germ cell in follicle. Scale bar: 20 μm. (**B**) Representative images of the G3BP2 of ovarian sections from E16.5 embryos, and PD0 and PD3 pups. Germ cells were marked by MVH (red); G3BP2 was stained with green; and nucleus was counterstained with Hoechst 33342 (blue). Dotted white line label the germline nest and white arrow mark the germ cell in follicle. Scale bar: 20 μm. (**C**) Representative images of the OOEP staining of ovary sections from E16.5 embryos, and PD0 and PD3 pups. Germ cells were marked by MVH (red), OOEP was stained with green and nucleus was counterstained with Hoechst 33342 (blue). Dotted white line label the germline nest and white arrow mark the germ cell in follicle. Scale bar: 20 μm. (**D**) Detection of G3BP2, EIF4A1, OOEP, ACTIN, and MVH proteins of E16.5 embryos, and PD0 and PD3 pups by western blot. ACTIN (only for MVH expression), and MVH was used as a loading control. Data were represented with mean ± SD ($n$ = 3 for independent repeats). The relative expression level was calculated between PD0 or PD3 and E16.5. Unpaired $t$ tests are performed. Statistical significance is shown as $^*P < 0.05$; $^{**}P < 0.01$. The raw data image can be found in S1 Data. The raw data used for quantification of D can be found in S2 Data. EIF4A1, eukaryotic translation initiation factor 4A1; E16.5, embryonic day 16.5; G3BP2, GTPase-activating protein-binding protein 2; HSPB11, heat shock protein family B member 11; MVH, mouse vasa homologue; OOEP, oocyte expressed protein; PD0, postnatal day 0; PD3, postnatal day 3.

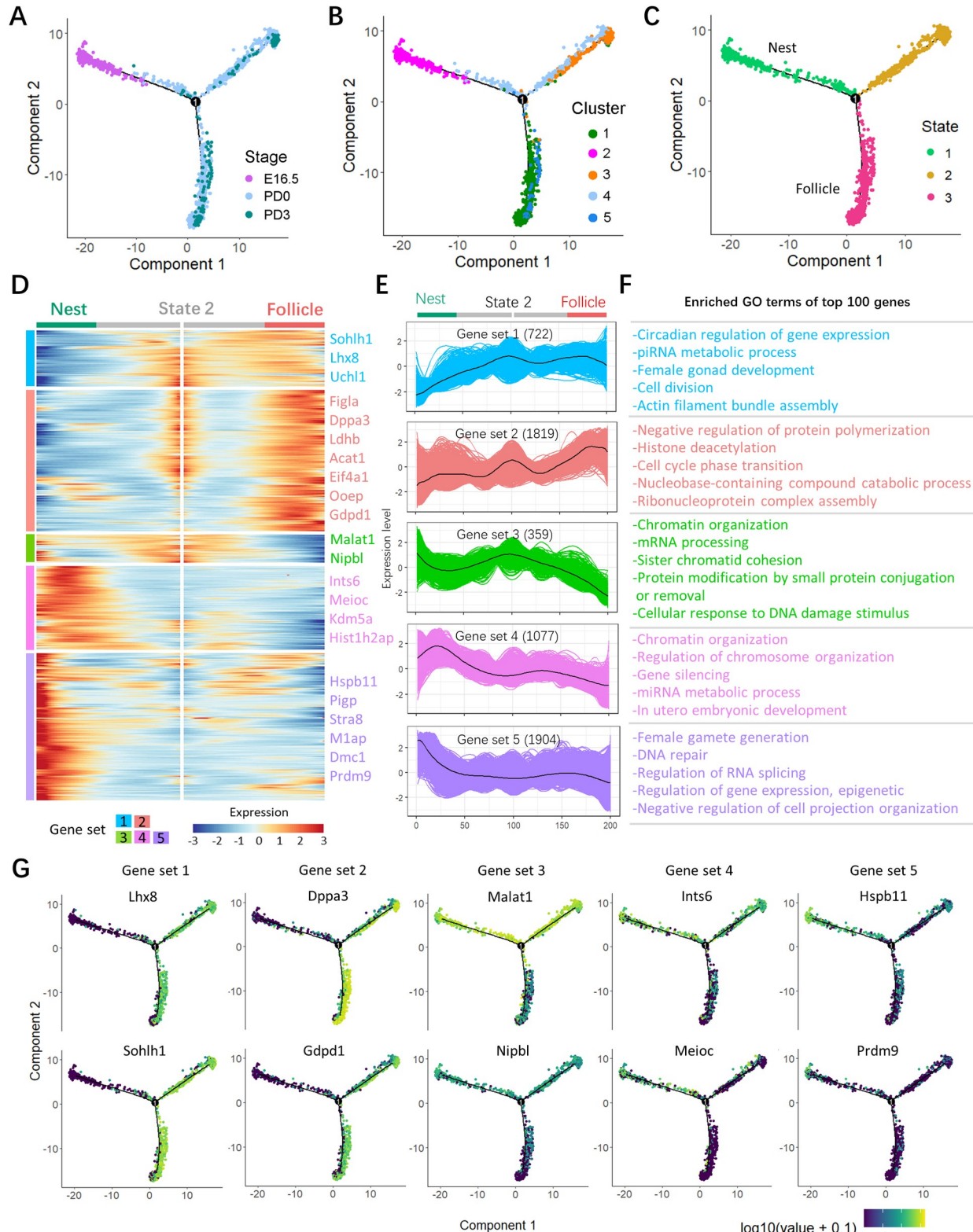

**Fig 4. Transition of germ cells throughout distinct states from nest to follicle stages.** (**A**) Single-cell trajectories of germ cell as a function of developmental timeline. (**B**) Single-cell trajectories of germ cells along with Seurat cluster. (**C**) Single-cell trajectories of the 3 germ cell states through the pseudotime. (**D**) Heatmap representing the expression dynamics of 5 gene sets with increased expression or reduced at the nest and follicle stage. The gene sets were selected with qval < $1 \times 10\mathrm{e}^{-4}$. Detailed information in gene sets is shown in S3 Table. (**E**) Expression profiles of

the differentially expressed genes in the transition from nest to follicle in each set. (**F**) GO terms of top 100 differentially expressed genes in each set. More GO terms are shown in S4 Table. (**G**) Expression of representative genes in each set along single-cell pseudotime trajectories. The sequencing data was deposited availably in GSE134339, and this figure can be produced using scripts at https://github.com/WangLab401/2020scRNA_murine_ovaries. GO, Gene Ontology.

### Gene expression signatures of the granulosa cell lineage

When the transcriptome of granulosa cells was plotted along with the developmental stages, 6 cell clusters were identified (Fig 6A). As noted, novel marker genes within each cluster were identified (S6 Table), whose expressions varied according to the developmental stages (S4A and S4B Fig). For example, *Wnt4* expression was most activated at E16.5, which may imply the differentiation of ovarian pre-granulosa cell and preparation of the second wave of follicle formation [19, 40]. Another, which showed a dramatic increase postnatally was *Aldh1a1*, a stem cell marker [41, 42], reported to be essential for the biosynthesis of retinoic acid (RA, a key regulatory factor of meiosis) [43]. At PD3, *Serpine2* is a gene encoding a member of the serpin family of proteins, a group of proteins that inhibit serine proteases and are also expressed by granulosa cells in human adult ovary [44].

Cell pseudotime trajectories revealed 3 granulosa cell states and 1 branch point (Fig 6B). During mouse ovary development, 2 waves of granulosa cell differentiation occur, characterized by partly distinct gene expression [15]; therefore, on the basis of the gene expression reported in Fig 6C and 6D, we attributed state 1 cells to early pre-granulosa cells expressing *Foxl2* and its downstream gene *Cdkn1c* [45, 46]. At state 2, another differentiating pre-granulosa cells expressed *Lgr5*, *Gng13*, and *Krt19*, and at state 3, the pre-granulosa cells expressed *Hsd3b1*, *Aard*, and *Akr1c14*, as well as *Foxl2*. These results suggested that state 1 cells were early bipotential pre-granulosa cells (early BPGs); that state 2 cells were differentiating epithelial pre-granulosa cells (D-EPGs); and that state 3 cells were differentiating BPGs (D-BPGs) [47]. For further characterization of the granulosa cell lineage, the differentially expressed genes of the 3 states were divided in 4 clusters according to their distinct expression patterns, and a heatmap was generated (Fig 6E, S7 Table). Gene sets 1 (460 genes) and 2 (722 genes) were assigned to state 3 (D-BPGs); set 3 (687 genes), to state 1 (early BPGs); and set 4 (1331 genes), to state 2 (D-EPGs). On this basis, *Cdkn1c* (set 3) expression decreased from early BPGs to D-BPGs; *Upk3b*, *Krt19*, and *Aldh1a2* (set 4) were high in D-EPGs and declined in D-BPGs; and *Aard*, *Hsd17b1*, *Inha*, and *Hsd3b1* (sets 1 and 2) were present at low levels in D-EPGs and high levels in D-BPGs (Fig 6E). Moreover, the expression dynamics of representative genes in each set are showed in Fig 6F. Furthermore, GO analysis of the enriched top 100 genes in each set was performed (Fig 6G). Gene set 1 was characterized by the expression of genes associated with biological processes, such as "regulation of cellular response to growth factor stimulus," "positive regulation of apoptotic process," and "negative regulation of cell differentiation," that were more highly expressed in D-BPGs and early BPGs than D-EPGs. Gene coding members of "c21-steroid hormone metabolic process" were among of set 2, confirming that c21-steroid hormones likely play a relevant role in PF formation. Whereas set 3 was composed of genes encoding proteins involved in "mitotic cell cycle process, muscle structure development, and epithelial cell proliferation," which both highly expressed in early BPGs and D-EPGs, set 4 consisted of genes enriched in "epithelial cell proliferation, endothelial cell migration, and cell–cell adhesion", which were more highly expressed in D-EPGs (S8 Table). In line with this, when sets 1 and 2 genes were analyzed by KEGG, they were highly enriched in "focal adhesion, oxytocin, and PPAR signaling" and cytochrome P450 related terms, as well as "ovarian steroidogenesis" in BPGs (S4C Fig). KEGG performed on sets 3 and 4 showed that the most represented terms of signaling pathways in EPGs were those of "PI3K-Akt, MAPK,

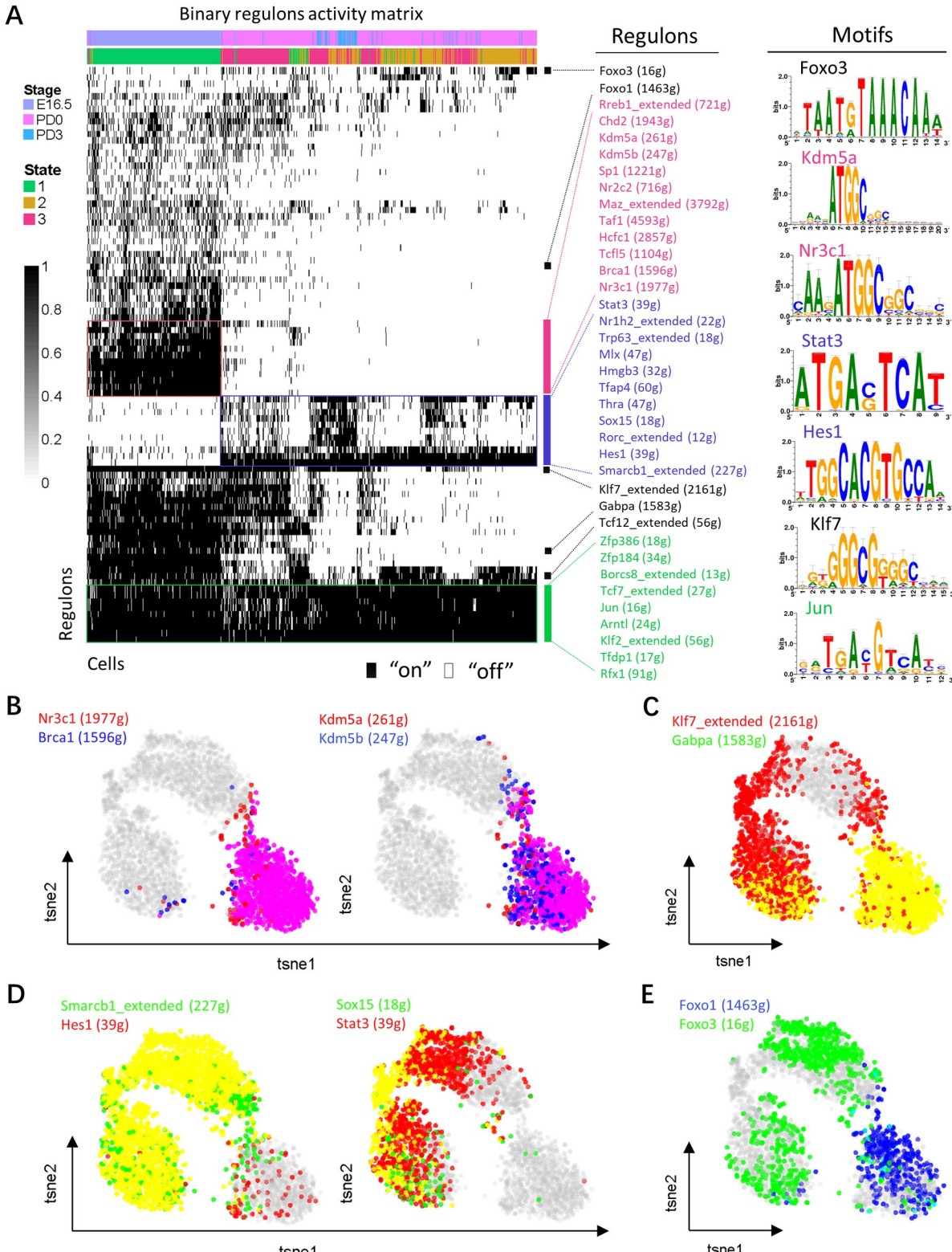

**Fig 5. Oocyte transcription factors involved in follicle assembly.** (**A**) Heatmap of regulon activity analyzed by SCENIC with default thresholds for binarization. The "regulon" refers to regulatory network of TFs and their target genes. "On" indicates active regulons; "Off" indicates inactive regulons. The top rows represent the transcriptome signature of selected germ cells, and they refer to the time point label and state character of Monocle (left panel). The representative TFs and part of their motifs involved in follicle assembly were listed in the

right panel. The numbers in parenthesis next to the regulon names indicate the number of genes enriched in regulons. (**B**) t-SNE projection of average binary regulon activity of *Nr3c1*, *Brca1*, *Kdm5a* and *Kdm5b* that were stage specific in germ cells in nests. (**C**) t-SNE projection of average binary regulon activity of *Klf7* and *Gabpa* that were most active in germ cells within nests. (**D**) t-SNE projection of average binary regulon activity of *Smarcb1*, *Hes1*, *Sox15*, and *Stat3* that were specific in follicular oocytes. (**E**) t-SNE projection of average binary regulon activity of *Foxo1* and *Foxo3* from the nest stage to follicle stage in germ cells. The sequencing data was deposited availably in GSE134339, and this figure can be produced using scripts at https://github.com/WangLab401/2020scRNA_murine_ovaries. SCENIC, single-cell regulatory network inference and clustering; TF, transcriptional factor; t-SNE, t-distributed stochastic neighbor embedding.

and Hippo" and to a lesser extent of "Rap1" (a Ras-related protein), while "ribosome, focal adhesions, tight junctions, and cell cycle" were among those more accounted for cellular functions (S4D Fig, S9 Table).

## Interactions between germ cells and granulosa cells during primordial follicle assembly

Since interactions between oocytes and granulosa cells are crucial during nest breakdown and PF assembly [35], and throughout the entire period of folliculogenesis, we compared the transcriptomes of germ cells and granulosa cells with each other according to the developmental stages, to verify the involvement of previously discovered players and identify new cell signaling pathways driving such interactions.

As seen in S5 Fig, and as previously reported, Vlnplots confirm that during the period of investigation, NOTCH signaling, members of the TGF-beta family, Kit/Kitl system and gap junction are important components of the oocyte–granulosa cell interaction [15,35]. In this regard, we found that genes encoding NOTCH ligands such as *Dll3* and *Jag1-2* were expressed by germ cells mainly at PD0 and/or PD3, respectively, while that for its receptor *Notch2* showed persistent expression in granulosa cells. The NOTCH-targeted gene *Hes1* showed transient expression in oocytes at PD0 and PD3 and resulted in ubiquitous expression in granulosa cells at all stages, whereas *Rbpj* was more highly expressed by germ cells at all stages and granulosa cells at PD0 and PD3 (S5A Fig). As expected, TGF-beta members, such as *Gdf9*, were mainly found in oocytes postnatally, while the gene encoding their receptors *Bmpr1a* and *Bmpr2* were expressed by oocytes in a paralleled manner and continually by granulosa cells. Genes encoding the transducers homologues of *Drosophila* protein, mothers against decapentaplegic and *Caenorhabditis elegans* protein sma (SMADs) were examined, and *Smad2* were expressed in both postnatal germ cells and granulosa cells. At the same time, there was no *Smad3* expression and decreasing levels of *Smad5* in germ cells, while *Smad3* and *Smad5* were found in granulosa cells throughout the investigated period. Finally, target *Id* genes were ubiquitously expressed at all stages both by germ cells and granulosa cells (S5B Fig). Again, as expected, *Kit* was similarly expressed by oocytes after birth, and *Kitl*, by granulosa cells at all stages (S5C Fig). Lastly, *Gjc1* and *Gja1* were found in oocytes and granulosa cells, respectively (S5D Fig), which encode gap junction proteins CX45 and CX43, supporting the participation of gap junction into the oocyte–granulosa cell interaction during the entire period of the study.

KEGG-enriched analyses were used to identify ongoing bioprocesses mainly in germ cells or granulosa cells or both (S10 Table). For example, "ribosome and RNA transport," followed by "autophagy-animal, cell cycle, and oocyte meiosis," were primarily found in germ cells; whereas "PI3K-AKT, MAPK, Rap1, and Hippo signaling pathways," followed by "focal adhesion, tight junction, and cell cycle," were mostly represented in granulosa cells (Fig 7A and 7B). Eight out of 25 processes found in both cell types referred to reproductive processes according to previous works [15,24] (S6A and S6B Fig), including "oocyte meiosis, cell cycle,

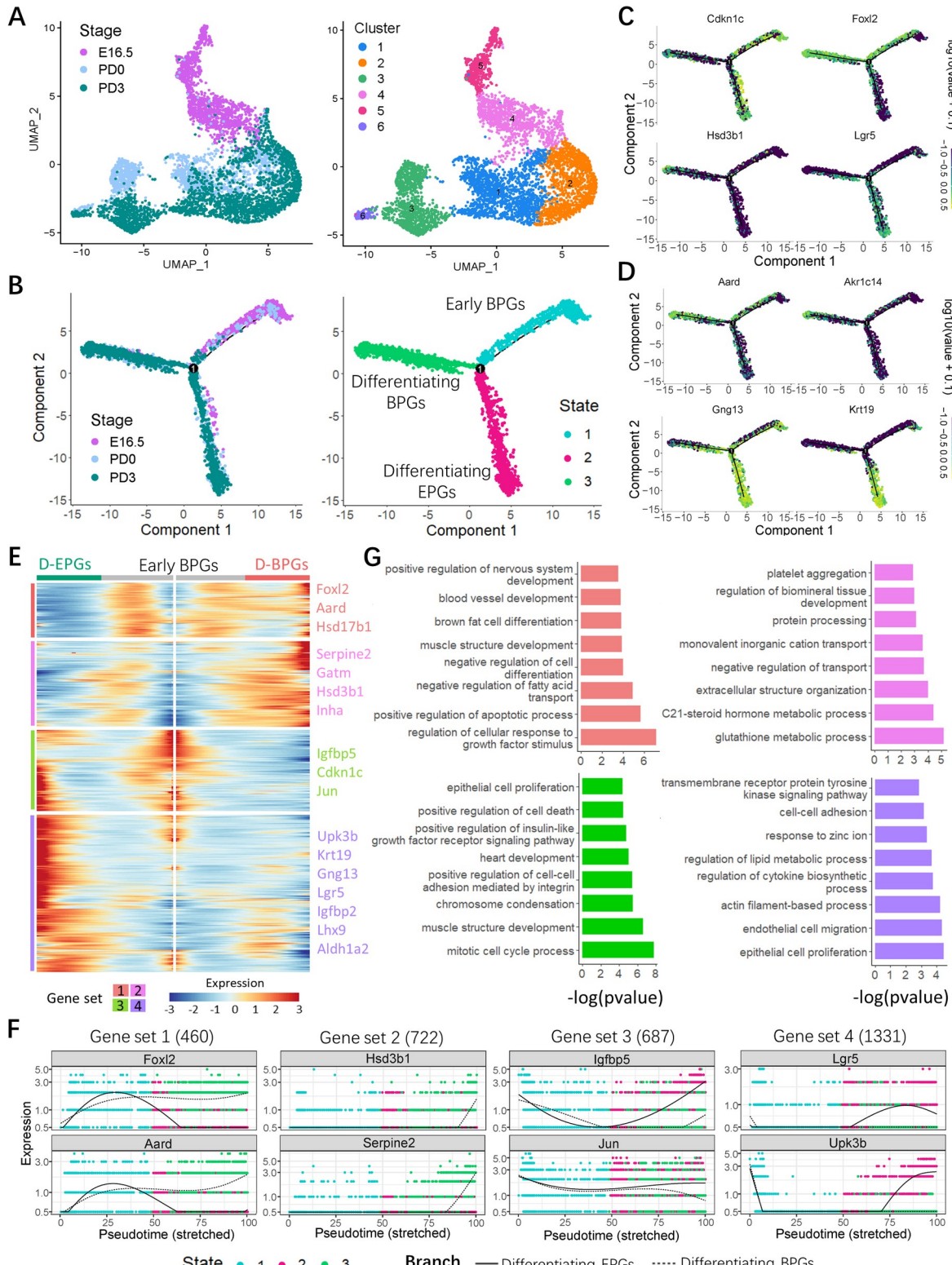

**Fig 6. Molecular characterization of granulosa cell subsets.** (**A**) Cluster analysis of granulosa cells with UMAP plots based on developmental timeline (left) and cell clusters (right). (**B**) Single-cell trajectories of granulosa cell subsets as a function of developmental timeline and cell States. (**C**) Expression of developmental marker genes along with pseudotime trajectories. (**D**) Expression of *Aard*, *Akr1c14*, *Gng13*, and *Krt19* along with pseudotime trajectories. (**E**) Heatmap of the gene expression programs of pre-granulosa cell with

different states. The gene sets were selected with qval $< 1 \times 10^{e-4}$. Detail information in gene sets is shown in S7 Table. Four gene sets represent increases or decreases in differentially expressed genes; part of the representative genes is listed in the right panel. "D-BPGs" refers to "Differentiating BPGs" and "D-EPGs" refers to "Differentiating EPGs". (F) Expression of representative genes in each set with 2 branches of pre-granulosa cell. (G) Enriched GO terms of the top 100 genes of each set. More GO terms are shown in S8 Table. The sequencing data was deposited availably in GSE134339, and this figure can be produced using scripts at https://github.com/WangLab401/2020scRNA_murine_ovaries. BPGs, bipotential pre-granulosa cells; D-BPGs, differentiating BPGs; D-EPGs, differentiating EPGs; EPGs, epithelial pre-granulosa cells; GO, Gene Ontology; UMAP, uniform manifold approximation projection.

adherens junction, and tight junction," and signaling pathways of "FoxO, Hippo, estrogen and p53."

To characterize the crosstalk between the 2 cell types, germ cells and granulosa cell transcriptome clusters were combined. Concerning FoxO signaling, *Foxo1* and *Foxo3* were the most common transcripts mainly expressed postnatally in oocytes and to a lesser extent in granulosa cells (Fig 7C–7E). Among FoxO downstream genes involved in cell proliferation (target 1), *Cdkn1b* was expressed at a higher level in granulosa cells at PD3, while autophagy-related genes (target 2), such as *Gabarapl 2*, *Map1lc3b*, and *Bnip3*, were expressed in both cell types throughout the examined stages although at different levels (Fig 7C–7E). As for Hippo signaling (Fig 7F), its key effectors, both *Yap1* and *Taz*, were more highly expressed in granulosa cells at all stages than in postnatal germ cells. For its downstream regulators, only *Sav1* had higher levels in germ cells, others, such as *Lats 1* and *2*, were more highly expressed in granulosa cells postnatally; however, its target *Cyr61* is most expressed at E16.5 (Fig 7G and 7H). Moreover, the expressions of FOXO3 and TAZ in germ cells were confirmed at the protein level by immunohistochemical staining (Fig 7I), and they were mainly expressed at the postnatal stage, which verified the data analysis results. Moreover, the transcripts for tight junction were expressed at the postnatal stage in both granulosa cells and germ cells (S6C and S6D Fig), while those for adherens junction proteins appeared to follow constantly disparate patterns, and they were present in abundance throughout almost all the timeline (S6E and S6F Fig).

## Transcriptional signatures of ovarian somatic cells during primordial follicle assembly

Specially, the transcriptional signatures of ovarian somatic cell types were further explored. As shown in S7A Fig, the stromal cell line was clustered, and 5 clusters were produced, which all expressed the specific marker genes of stromal cells (S7B Fig), while the novel marker genes in each cell cluster were identified (S7C Fig). Additionally, the cell clustering analyses of endothelial cells (S7D Fig) and immune cells (S7E Fig) were performed, and 3 and 4 clusters were generated, respectively. More notably, compared with transcriptome patterns of stromal and endothelial cells, whose transcriptome indicated the distinct patterns dependent on pre- and postnatal manners, the transcriptional signature of immune cells was distinguished from them both, and it seemed to be more irregular. However, the potential roles of ovarian somatic cells participating in PF assembly still need to be addressed.

## Discussion

In the present study, we used scRNA-seq analyses to identify the complete transcriptome programs of the different cell types present in mouse ovaries from the end of fetal age through the first few days after birth. As reported in the Introduction, in most mammalian species, this period is crucial for ovarian development and is characterized by several processes involving all ovarian cell populations. In this regard, our analyses were able to identify 7 distinct cell types, including germ cells, granulosa cells, stromal cells, epithelial cells, erythrocytes, immune

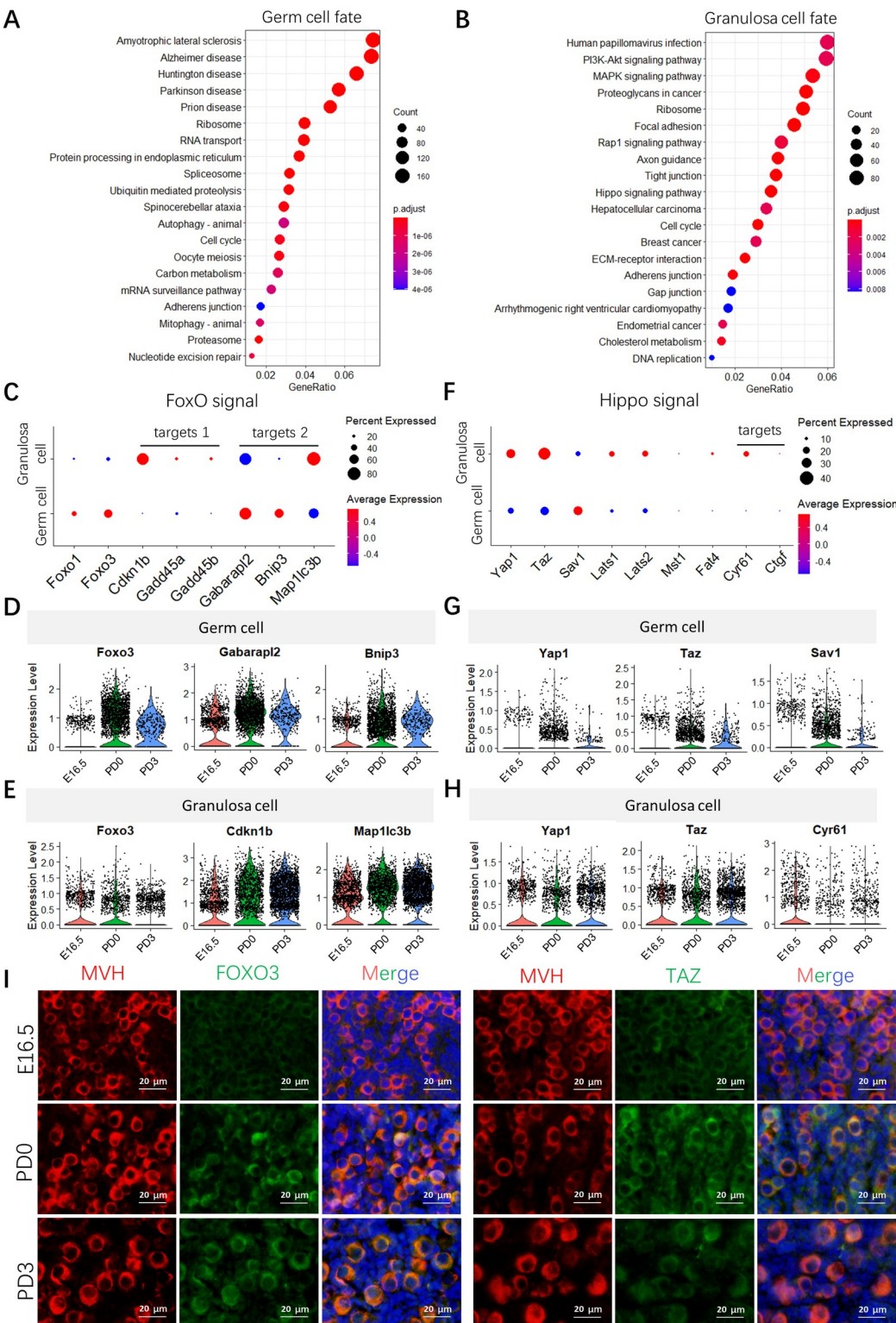

**Fig 7. Enrichment molecular pathways of oocytes and granulosa cell subsets during follicle assembly.** (**A** and **B**) Pathway enrichment of key genes regard to fate transitions of germ cells (**A**) (total of 5,881 genes at branch point) and granulosa cell (**B**) (total 3,200 genes at branch point). More pathways are shown in S10 Table. (**C**) Dot plots of FoxO signal and its target genes in germ cell and granulosa cell clusters. (**D** and **E**) Expression of representative genes of the FoxO signal in germ cells (**D**) and granulosa cells (**E**). (**F**) Dot plots of the Hippo signal in germ cells and granulosa cells. (**G** and **H**)

Expression of representative genes of the Hippo signal in germ cells (**G**) and granulosa cells (**H**). (**I**) Representative images of FOXO3 and TAZ immunohistochemistry staining of ovarian sections from E16.5 embryos and PD0 and PD3 pups. Germ cells were marked by MVH (red); FOXO3 and TAZ were stained with green; and nucleus was counterstained with Hoechst 33342 (blue). Scale bar: 20 μm. The sequencing data was deposited availably in GSE134339, and this figure can be produced using scripts at https://github.com/WangLab401/2020scRNA_murine_ovaries. E, embryonic day; FOXO3, Forkhead box O3; PD, postnatal day; TAZ, tafazzin family protein.

cells, and endothelial cells that underwent dynamic changes throughout the period of investigation (Fig 1B–1E). These changes likely reflect the dynamics of proliferation, differentiation, and death of the different cell populations that, with regards to germ cells, are characterized by nest breakdown associated with extensive degeneration. In fact, some ovarian somatic cells, such as stromal (S7A–S7C Fig), endothelial cells (S7D Fig), and immune cells (S7E Fig); however, due to the limited information available regarding other ovarian somatic compartments, the current study placed most attention on germ cells and granulosa cells.

Based on the expression of known cell lineage and developmental stage-specific genes, we generated 5 different cell clusters for germ cells. According to this analysis, a genetic dynamic model, including pre-, early- and late-follicle formation stages of germ cell development, and the changes of transcriptome signature in germ cell at these stages were delineated (Fig 2A–2D). Genes known to be related to meiotic processes, such as *Prdm9* [31], showed widespread high expression in cluster 2, as well as *Stra8*, encoding a protein crucial for the beginning of meiosis [48], only as expected since its expression was reported to be limited to the early stages of meiotic prophase 1 [49]. With the exception of *Figlα* that was expressed at variable levels in all clusters, also other genes, such as *Lhx8*, *Nobox*, and *Sohlh1* (encoding transcription factors reported to be typical of follicular oocytes [15]), appeared restricted to a few clusters (1, 3, and 4), and assigned to the early follicle stage. Finally, *Gdf9* (encoding a growth factor of the TGF-beta family [50]), *Zp2*, and *Zp3* (encoding zona pellucida proteins [35,36]) showed an even more restricted expression that was limited mainly to cluster 5, corresponding to the late follicle formation stage. All these gene dynamics potentially play an important role in PF formation (upper panel in Fig 8).

Overall, the above results indicated the validity of our analyses and made us confident that we had identified several new genes expressed by germ cells at the 3 designated stages, whose importance and role can be the focus of future studies. For example, at the pre-follicle stage, *M1ap* and *Hspb11*, a protein required for meiosis I arrest during spermatogenesis [51] and a heat shock protein that inhibits cell death through stabilization of the mitochondrial membrane [52], respectively, are likely candidates for a role in the control of female germ cell meiosis and in their survival/apoptosis at the pre-follicle stage. At the early follicle stage, the expression of the following genes suggest an increase of both general and specific transcriptions in oocytes: *Eif4a1*, encoding an ATP-dependent RNA helicase, a subunit of the eIF4F complex involved in cap recognition and required for mRNA binding to ribosomes [53]; *G3bp2*, encoding an ATP- and magnesium-dependent helicase, a member of nuclear RNA-binding proteins [54], and *Id1*, encoding a helix-loop-helix (HLH) protein forming heterodimers with the basic HLH family of transcription factors that may play a role in cell growth, senescence, and differentiation [55]. At the same time, the expression of 2 genes: *Ooep* (which encodes an RNA binding protein of the subcortical maternal complex (SCMC) of the oocyte and likely participates oocyte homologous recombination [56,57]) and *Padi6* (which encodes a member of the peptidyl arginine deiminase family of enzymes that may play a role in cytoskeleton reorganization in the egg and in early embryo development [58–60]) both support findings that the oocytes begin to express these proteins at the late stage of PF assembly. It is important to note that the experimental approach used in the present paper identifies potential

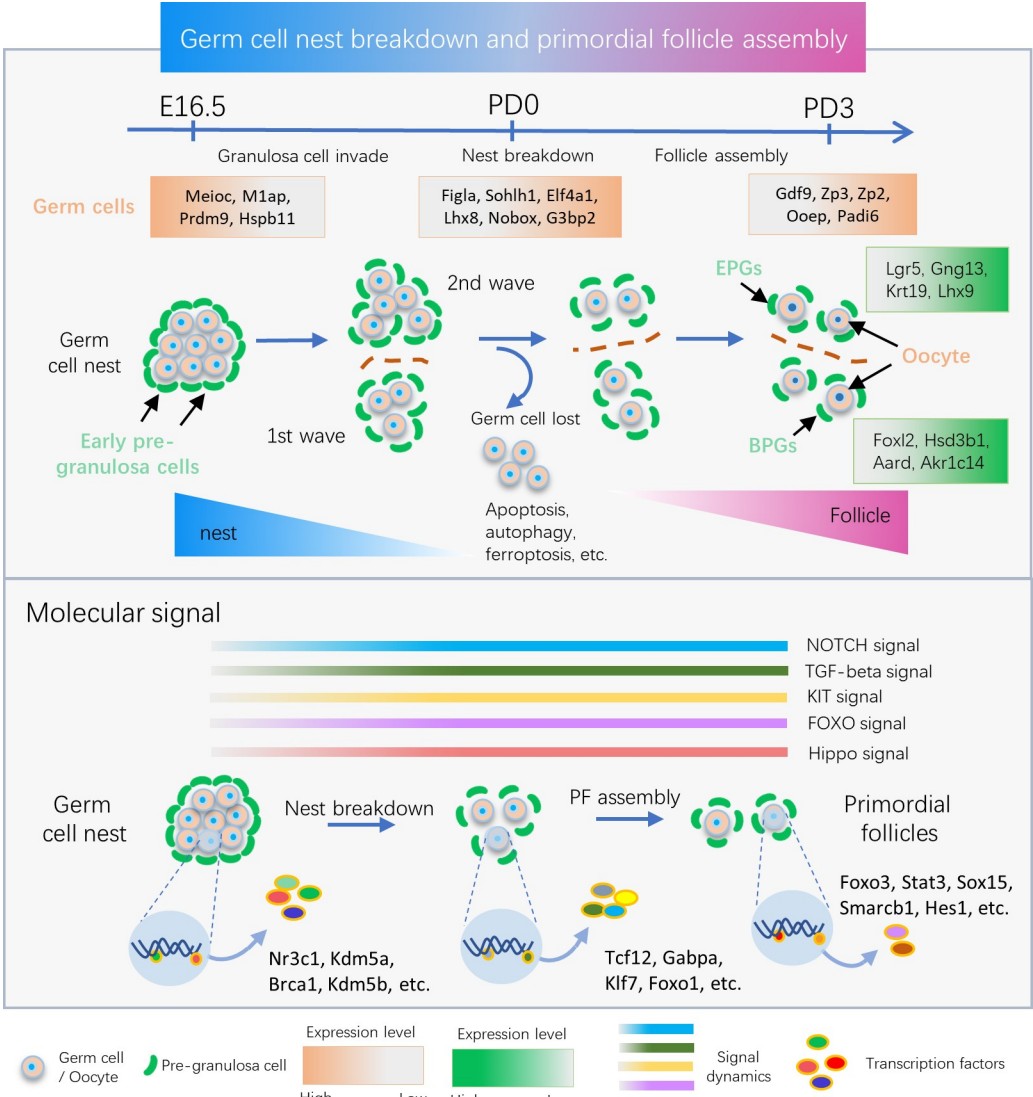

**Fig 8. Schematic scheme of the main genetic and molecular signal dynamics detected in ovarian cells during follicle assembly.** The upper panel indicates the genetic programs and key biological events of germ cells and pre-granulosa cells during germline nest breakdown and follicle assembly, and the developmental time courses are indicated above; the orange box contains the marker genes and their expression dynamics in germ cell; the green ones comprise the marker genes and their expression dynamics in pre-granulosa cell. The dark orange dashed line separates ovarian cortex (above) from the medulla, labeled with 2 sources of pre-granulosa cells, i.e., BPG cells for the first wave in ovarian medulla and EPG cells in ovarian cortex for the second wave of follicle formation, respectively. In the bottom panel (upper part), the activation of 5 main interactive pathways between germ cells and granulosa cells during these processes is represented. In the lower part, the stage-specific TFs in oocytes from nest to follicle stage are reported. BPG, bipotential pre-granulosa; EPG, epithelial pre-granulosa; TF, transcriptional factor.

candidate genes involved in the investigated events (upper panel in Fig 8). Further analysis at an individual gene level is required to determine the specific function of a given gene product.

By ordering germ cells along pseudotime trajectories, we were able to reconstruct the gene dynamics involved in the transition of oocytes from the pre-follicle stage (mostly germ cells in nests) to the follicle stage (PF assembly) throughout 3 distinct differentiation states. The majority cells at pre-follicle stage belonged to state 1, most of which were germ cells in nests, while those at early- and late-follicle stages were in states 2 and 3 and were considered as

follicular oocytes. Five gene sets were identified including those genes likely involved in the commitment of germ cells present in nests into follicular oocytes. Among these, the most differentially expressed were *Sohlh1*, *Lhx8*, and *Uchl1* (set 1) and *Figla*, *Dppa3*, *Eif4a1*, and *Ooep* (set 2), with moderately and highly increased expression, respectively, in early and late follicular oocytes. *Malat1* and *Nipbl* (set 3) were highly expressed in germ cells in nests and early-follicle formation oocytes; *Ints6*, *Meioc*, *Kdm5a*, and *Hist1h2ap* (set 4) were dramatically upregulated in germ cells in nest. Enriched GO applied to the 100 most highly expressed genes of these gene sets involved in specific biological process (Fig 4F). Of note, the GO terms of "histone deacetylation" and "regulation of gene expression and epigenetic" were enriched, and these results indicated the epigenetic modification of germ cell likely involved fate decisions of its own during follicle formation. Given the fact of germ cell loss, apoptosis has been the most commonly identified process; however, the present scRNA-seq platform does not excel in the determination of apoptotic cells, and dying cells are regarded as low quality and therefore needed to be discarded; another known is mechanism of cell autophagy, which was enriched (Fig 7A); it is worth noting that ferroptosis, 1 form of cell death, was revealed in the current study (S10 Table), and its roles in oocyte loss deserves to be investigated further.

Moreover, using the SCENIC algorithm, we were able to establish a network of regulons, that can be postulated as likely candidates for sustaining germ cell specific TF programs throughout the period of investigation (lower panel in Fig 8). Through using this method, we revealed several TFs at distinct stages, such as *Nr3c1*, *Foxo1*, *Brca1*, *Kdm5a*, and *Kdm5b* at the germ cell nest stage; *Smarcb1*, *Hes1*, *Stat3*, and *Sox15* during the period of PF assembly; and *Foxo3* mainly limited postnatal period (Fig 5). It is worth noting that *Nr3c1* has been revealed as the meiotic specific TF in oocyte; furthermore, *Brca1*, *Kdm5a*, and *Kdm5b* have previously been detected in meiotic oocyte [61]. Meanwhile, recent studies have shown that FOXO3 plays a critical role in the maintenance of PF dormancy and *Foxo3* knockout lead to follicles overactivation [62–64], and the results indicate *Foxo3* is a suppressor of PF activation and may be required for their formation. In addition, SMARCB1 is essential for embryo development [65]; however, its role in PF has been unknown, which need further investigated.

Bioinformatics analyses of the granulosa cell transcriptome identified 6 clusters and 3 states (Fig 7B). As for the genes known to be expressed by the granulosa lineage during the period of investigation, it showed that some of these genes, such as *Foxl2* and *Lgr5*, varied considerably at different stages. Moreover, novel genes within each cluster were identified, whose expressions changed along with the developmental stages. Cell pseudotime trajectories revealed 3 granulosa cell states and 1 branch point (Fig 6B and 6C). Following the considerations reported in the results, state 1 was attributed to early pre-granulosa cells, and it branched at state 2 of pre-granulosa cells expressing *Lgr5* and *Gng13* and at state 3 of late stage pre-granulosa cells expressing *Foxl2* and *Hsd17b1* that are required for steroidogenesis [30]. Actually, the origin of granulosa cell is always indefinite, which was hypothesized to be derived from mesonephros or ovarian surface epithelial cells [66]. As recently reported [47], there are 2 granulosa cell differentiation pathways supporting follicle formation, which are marked by *Foxl2* and *Lgr5* expressions, and they show different transcriptional patterns and are involved in 2 waves of follicle formation. In our results, the 2 distinct cell populations above were indeed confirmed, and analogously, state 3 cells with high expression of *Foxl2*, *Aard*, and *Akr1c14* are suggested to be BPGs supporting follicle in the ovarian medulla; meanwhile, the EPGs are for oocytes destined for the ovarian cortex, as evidenced by the expression of *Lgr5*, *Gng13*, and *Krt19* at state 2 (Fig 6D). Thus, the consistent results in 2 studies further evidenced the notion that 2 kinds of pre-granulosa cells supporting follicle formation respectively at ovarian medulla and cortex in mice.

Further, bioinformatics analyses of germ cell and granulosa cell transcriptomes paralleled in function with the developmental stages confirmed that NOTCH signaling, members of the TGF-beta family, Kit/Kitl system and gap junctions, are important components of the oocyte–granulosa cell interactions during the period of investigation (lower panel in Fig 8). In addition, it showed that ligands, receptors, and signaling mediators of these system were coherently expressed mainly in germ cells, pre-granulosa cells, or both, as detailed in the results. Moreover, basically in line with the finding discussed above, biological processes, such as "RNA transport, cell cycle, and meiosis," were primarily found in germ cells, whereas "proteoglycan and cholesterol metabolism" were more commonly represented in granulosa cells and "ribosome biogenesis" in both cell types. Hippo pathways were mostly represented in granulosa cells, and FoxO signaling were present in both cell types although at different developmental stages. The components of Hippo signaling in granulosa cells have been reported in several mammalian species [67–69]. Previous reports suggest Hippo kinases large tumor suppressor kinase (LATS) 1 and 2 are required for ovarian granulosa cell fate maintenance [70] and knockout of yes-associated protein 1 (Yap1) in granulosa cells block follicle development [71]. Furthermore, disruption of the Hippo pathway has been reported to promote AKT-stimulated ovarian follicle growth [72]. Here, we reported its possible involvement in the early stages of granulosa cell development. As for FoxO, while a higher level of transcripts for *Foxo1* and *Foxo3* was present in postnatal oocytes, *Cdkn1b* was expressed at a higher level in granulosa cells at PD3, and autophagy-related genes (*Gabarapl2*, *Map1lc3b*, and *Bnip3*) were expressed in both cell types throughout the examined stages although at different levels. This suggests that FoxO signaling is not only important in controlling the cell cycle [73] and survival of the postnatal oocytes as reported in previous studies [74,75], but is involved in granulosa cell proliferation and autophagy.

Furthermore, despite the fact that the single-cell transcriptome technique is greatly expanding our insights into biological systems, especially regarding cellular heterogeneity, there are some unsolved issues that still need addressing within the field. Firstly, it is the current lack of standardization for date analysis [76]. The problem is triggered by the growing number of analysis methods, exploding dataset sizes, and technical aspects (such as a variety of programming languages for data analysis tools), all of which prevent computational biologists from developing a gold standard practices. Another is regard to the incorporate of biological replicates [77,78]. Although the biological replicates are suggested to eliminate batch effects, it seems the 1 replicate is still more popular in the single-cell dataset of recent publications [47,61]. In the present study, the application of 1 replicate may be considered as a potential limitation; however, the canonical knowledge in the field of PF formation, as well as the experimental verification, makes us confident for our results. Moreover, the cell proportions studied here, at least partially, do not reflect the true cell type compositions in ovarian tissue (such as germ cells), and this may be attributed to the methods of tissue handling and dissociation for single-cell sample preparation, which is also present in another publication of single cell transcriptome investigating the female ovary [79].

In summary, the present study highlighted the transcriptional dynamics taking place in oocytes and granulosa cells during follicular formation by scRNA-seq. Specifically, the development stage-specific marker genes of germ cells and granulosa cells are identified. Furthermore, the TF regulatory networks of germ cells and novel pathways involved in PF formation have been given preliminarily confirmation. Thus, the current study provides novel insights into follicle formation at the perinatal stage and offers a reliable resource for study of the molecular regulatory mechanism of follicular development in the mouse.

## Materials and methods

### Animals and preparation of cell suspensions

C57BL/6J strain mice were purchased for scRNA-seq and ovarian sections staining from Vital River Laboratory Animal Technology Co. (Beijing, China). CD-1 mice for western blot experiments only were purchased from Jinan Pengyue Experimental Animal Breeding Co. Ltd (Jinan, China). Females were mated with males in the afternoon, and the presence of a vaginal plug in the morning of the following day was considered as E0.5. Embryonic and postnatal ovaries were dissected from E16.5 embryos and pups at PD0 and PD3, respectively. The selected 3 time points for sequencing represent the key stage of a series of cellular events involving PF formation, i.e., E16.5 time point represents the germ cell preparation (see prerequisites cellular events for the establishment of PFs [15]), and PD0 is the initial time point of nest breakdown, when it comes to PD3, more PFs have formed.

To obtain single-cell populations, isolated ovaries were cut into small pieces and incubated in 0.25% trypsin (Hyclone, Beijing, China) and collagenase (2 mg/ml, Sigma-Aldrich, C5138, Shanghai, China) for 6 to 8 min at 37˚C. Tissues were disaggregated with a pipette to generate single cells, and the solution was filtered through 40-μm cell strainers (BD Falcon, 352340, United States of America) and washed 2 times with PBS containing 0.04% BSA. Cell viability was acceptable when after staining in 0.4% Trypan Blue; it was above 80%; and cell concentration (1,000 cells/μL) was checked to meet the requirement of sequencing, as well as single-cell rate (no connected cells observed during cell counting).

### Ethics statement

Animals were housed according to the national guideline, the mice were humanely killed by cervical dislocation, and other experimental operations on mice have been approved by Animal Care and Ethical Committee of Qingdao Agricultural University.

### Single-cell libraries and sequencing

With qualified single cell sample, cells in 0.04% BSA in PBS were loaded onto 10× Chromium chip, and single-cell gel beads in emulsion were generated by using Single Cell 3′ Library and Gel Bead Kit V2 (10× Genomics Inc., 120237, Pleasanton, California, USA). Followed the manufacturer's instructions, single-cell RNA-seq libraries were constructed, and pair-end 150 bp sequencing was performed to produce high-quantity data on an Illumina HiSeq X Ten (Illumina, San Diego, California, USA). Further, the "mkfastq" module of Cell Ranger produced the FASTQ files with raw base call (BCL) files as input generated by Illumina sequencer alignment.

### Clustering analysis with Seurat and cell trajectory construction by Monocle

The Cell Ranger "count" pipeline (version 3.1.0) were applied with the produced FASTQ data to map the mouse reference genome (version mm10) and process feature–barcode matrices with default parameters, specially, to avoid potential bias during subsequent bioinformatics analysis, the captured cell of each sample in number is set to 6,000 with the parameter of "—force-cells." The data matrixes were then loaded in R (version 3.6) using the Seurat package (version 3.1.5) [80]. The Seurat object was created based on 2 filtering parameters of "min. cells = 5" and "low.thresholds = 200," and the exorbitant number of unique genes detected in each cell (i.e., "nFeature_RNA") was adjusted in each sample to eliminate the empty drop and dying cell and also potential doublets/multiplets for subsequent analyses (S1C Fig). Moreover, the second check was performed in single-cell type to guarantee high-quality cells, including

the distribution of detected unique genes (cells with an over-high number of unique genes are viewed as potential doublets/multiplets) and abnormal gene expression in each cell type (for example, if the germ cell specifically expressed marker genes of other cell types, these cells would undergo abnormal gene expression and be discarded). Then, the multiple samples were performed an integrated analysis, and briefly, the "FindIntegrationAnchors" function was used to identify anchors (a strategy for identification of cell pairwise correspondences between single cells across datasets, termed "anchors," which can be used to transform datasets into a shared space and construct the harmonized atlases at the tissue or organismal scale) of 3 Seurat objects, and the integration analysis was performed according to anchors among samples [81]. Followed by the normalizing and scaling steps, with the UMAP (a visualized method for cell clustering in high-dimensional transcriptomic data) technique [82], a series of command was executed to visualize cell clusters, such as "RunUMAP" function with proper combination of "resolution" and "dims.use"; "FindNeighbors" and "FindClusters" function to conduct cell clustering. To identify canonical cell cluster marker genes, "FindAllMarkers" function was used to identify conserved marker gene in clusters with default parameters.

Further, the subpopulation of germ line or granulosa cell line was imported into Monocle (version 2.10.1) [83], a novel unsupervised algorithm, which reordered the cells to maximize the transcriptional similarity by their progress through differentiation, and at the same time, it would distinguish genes activated early and later in differentiation, which contribute to dissecting of cell differentiation fate, also termed "pseudotime analysis." With the gene count matrix as input, the new dataset for Monocle object was created, and functions of "reduceDimension" and "orderCells" were carried out to generate the cell trajectory based on pseudotime. Particularly, the ordering genes were differentially expressed genes between clusters in each cell type calculated by "differentialGeneTest" function in Monocle. In addition, the "BEAM" function was used to calculate the differentially expressed genes at a branch point in the trajectory, and a gene with "qval $< 1 \times 10^{e-4}$" showed with heatmap. Moreover, the root state (that is, a prebranch in the heatmap) was set and adjusted following consideration of the biological meanings of different cell branches.

## The regulon activity of transcription factors with SCENIC

The SCENIC algorithm had been developed to assess the regulatory network analysis regard to TFs and discover regulons (that is, TFs and their target genes) in individual cells. Following the standard pipeline, the gene expression matrix with gene names in rows and cells in columns was input to SCENIC (version 0.9.1) [39]. The genes were filtered with default parameter, and finally 9,208 genes were available in RcisTarget database, the mouse specific database (mm10) that is used as the default in SCENIC. The co-expressed genes for each TF were constructed with GENIE3 software, followed by Spearman's correlation between the TF and the potential targets, and then the "runSCENIC" procedure assisted to generate the GRNs (also termed regulons). Finally, regulon activity was analyzed by AUCell (Area Under the Curve) software, where a default threshold was applied to binarize the specific regulons ("0" present "off" of TFs, and "1" refer to "on"). The parameters of t-SNE was set with 50 principal components (PCs) and 50 perplexity to visualize (after running a consistency test across several perplexity values and number of PCs); meanwhile, the cell states were mapped with specific regulons, and averages of binary regulon activity (Fig 5B–5E) and TF expressions (S3C–S3E Fig) were projected onto t-SNEs.

## GO and KEGG enrichment analysis of gene set

For a large gene set with more than 100 genes, the enrichment analysis of GO and KEGG is preformatted with ClusterProfiler [84], a R package in Bioconductor, was applied to detect the

gene-related biological process, and signaling pathways with threshold value of "pvalueCut-off = 0.05," and top terms were displayed (Fig 7A and 7B and S2F, S2G, S4C and S4D Figs). The online tools Metascape is used to uncover the function GO terms of small gene sets [85] (Figs 4F and 6G).

## Immunohistochemistry

Immunohistochemistry was performed as previously described [86]. Ovarian samples were processed for paraffin inclusion after fixing with paraformaldehyde (Solarbio, P1110, China) and following a standard dehydration procedure. Sample sections of 5 μm in thickness were blocked at room temperature for 30 min after gradient rehydration and antigen retrieval and incubated with the germ cell-specific mouse vasa homologue (MVH) antibody (Abcam, ab13840, USA) overnight at 4°C. After 3 washes in TBS, the sections were incubated with secondary antibodies of FITC-labeled goat anti-rabbit IgG (H + L) (Beyotime, A0562, China) for 30 min at 37°C. After a further 3 washes in TBS, slides were counterstained with propidium iodide (PI, Sangon Biotech, E607306, China) for 3 min and sealed. Photomicrographs were taken with a fluorescence microscope (Olympus, BX51, Japan). Double immunostaining was performed with MVH antibody from mouse (Abcam, ab27591) or rabbit (Abcam, ab13840) with others: HSPB11 (Proteintech, 15732-1-AP, USA), G3BP2 (ABclonal, A6026, USA), OOEP (Biorbyt, orb379068, USA), FOXO3 (Novus Biological, NBP2-16521, USA), and TAZ (ab84927, Abcam) antibodies, followed by secondary antibodies of CY3-labeled goat anti-mouse IgG (H+L) (Beyotime, A0521) and goat anti-rabbit IgG H&L (Alexa Fluor 488, Abcam, ab150077); Hoechst 33342 or PI were used for nucleus counterstaining. In addition, the germ cells with more than 2 cells interconnected were considered as nest germ cells in ovarian sections, whereas single germ cell without connection to each other were considered as follicle germ cells; the percentage of germ cells within nests or follicle was calculated.

## Western blot

The western blot was administrated as previous reports [86, 87]. Briefly, with the collection of 6 ovaries, proteins in each sample were extracted using RIPA lysis solution (Beyotime, P0013C). After SDS-PAGE, the proteins were separated and transferred on polyvinylidene fluoride membrane (PVDF, Millipore, ISEQ00010, USA). They were then blocked with TBST buffer (TBS with 1% Tween-20) containing 5% bovine serum albumin, and the membrane was subsequently incubated with ACTIN (Sangon Biotech, D110007) or EIF4A1 antibody (A5294, ABclonal) (other primary antibodies see above) overnight at 4°C. The next day, after 3 washes, the membranes were incubated with secondary antibodies of HRP-conjugated goat anti-rabbit (Beyotime, A0208) or anti-mouse IgG (Beyotime, A0216) at room temperature for 1.5 h. Finally, a BeyoECL Plus Kit (Beyotime, A0018) was applied for chemiluminescence, according to the manufacturer's instructions. The expression of target protein was quantified by using the AlphaView SA software (ProteinSimple, California, USA).

## Statistical analysis

Results data as mean ± SD were obtained from 3 independent experiments. The statistical analysis was performed with GraphPad Prism software (version 5.0), and the significant difference was determined with 2-tailed student's unpaired $t$ test. Significant and highly significant difference were set at $P < 0.05$ and $P < 0.01$, respectively.

## Supporting information

**S1 Fig. Information of single cell sequencing and cell types identification in ovary.** (**A**) Representative images of fetal ovaries at E16.5 and postnatal ovaries at PD0 and PD3. Germ cells were labeled with MVH (green) and nuclei counterstained with PI (red). Scale bar: 20 μm. (**B**) Percentage of germ cells in nests or follicles at the indicated stages. Data were represented with mean ± SD (*n* = 3 for independent repeats). The relative level was calculated between PD0 or PD3 and E16.5. Unpaired *t* tests are performed. Statistical significance is shown as $^{**}$ *P* < 0.01; ns *P* >0.05. The raw data used for quantification of B can be found in S3 Data. (**C**) The sequenced detail information of 3 samples after CellRanger and Seurat workflow. (**D**) Feature plots of specific marker genes of erythrocytes, endothelial cells, immune cells, and epithelial cells. (**E**) Percentages of the 6 ovarian cell types at E16.5, PD0, and PD3. The sequencing data was deposited availably in GSE134339, and this figure can be produced using scripts at https://github.com/WangLab401/2020scRNA_murine_ovaries. E16.5, embryonic day 16.5; PD0, postnatal day 0; PD3, postnatal day 3; MVH, mouse vasa homologue; PI, prodium iodide.
(TIF)

**S2 Fig. Marker gene expression with developmental time course, cell proportion, and enrichment analysis in germ cells.** (**A**) Heatmap of top 10 marker genes of germ cell cluster with developmental timeline. Top 50 marker genes in each cluster are shown in S2 Table. (**B**) Percentage of germ cells at pre-, early- and late-follicle formation stages. (**C**) The expressions of representative genes for 3 identified stages along with pseudotime trajectories. (**D**) Expression of representative genes (*Hdac2* and *Morf4l1*) in GO term of "histone deacetylation" along with pseudotime trajectories. (**E**) Expression of representative genes (*Morc2a* and *Smarca4*) in GO term of "regulation of gene expression and epigenetic" along with pseudotime trajectories. (**F**) Pathway enrichment of highly un-regulated genes of germ cells in follicle. (**G**) Pathway enrichment of most expressed genes of germ cells in nest. More pathways are shown in S5 Table. GO, Gene Ontology.
(TIF)

**S3 Fig. The t-SNE projection map of cell clustering, regulons density, and TF expressions.** (**A**) t-SNE based on 201 regulons with 50 PCs and 50 perplexity according to the developmental stages (left) and cell states (right). (**B**) t-SNE of regulons activity density. Regulon density means "the occurrence frequency of regulon in cells." (**C**) TF expressions of *Nr3c1*, *Brca1* *Kdm5a*, and *Kdm5b* in selected cells. (**D**) TF expressions of *Smarcb1*, *Hes1*, *Stat3*, and *Sox15* in selected cells. (**E**) t-SNE projection of average binary regulon activity (left) of *Klf2* and *Jun* throughout the developmental stages and their expressions (right) in germ cells. TF, transcriptional factor; t-SNE, t-distributed stochastic neighbor embedding; PC, principal component.
(TIF)

**S4 Fig. Marker gene expression with cell cluster and with developmental time points and pathway enrichment analysis of pre-granulosa cells.** (**A**) Heatmap of the top 5 marker genes in granulosa cell clusters. Top 50 marker genes in each cluster are shown in S6 Table. (**B**) Vlnplots of the representative genes in granulosa cell clusters according to the developmental stages. (**C**) KEGG pathway enrichment of gene sets 1 and 2 that were related to BPGs fate. (**D**) KEGG pathway enrichment of gene sets 3 and 4 that have high expression in EPGs. More KEGG pathways are shown in S9 Table. The sequencing data was deposited availably in GSE134339, and this figure can be produced using scripts at https://github.com/WangLab401/2020scRNA_murine_ovaries. BPG, bipotential pre-granulosa; EPG, epithelial pre-granulosa; KEGG, Kyoto Encyclopedia of Genes and Genomes.
(TIF)

**S5 Fig. Interaction of germ cell and pre-granulosa cell mediated by typical signal pathways.**
(**A**) Vnlplots of the expression of NOTCH signal ligands, receptors, and targets in germ cells and granulosa cells. (**B**) Vnlplots of the expression of TGF-beta signal ligands, receptors, effectors, and targets in germ cells and granulosa cells. (**C**) Vnlplots of the expression of *Kit* and *Kitl* in germ cells and granulosa cells. (**D**) Vnlplots of the expression of connexin genes of gap junction in germ cells and granulosa cells. TGF-beta, transforming growth factor beta.
(TIF)

**S6 Fig. Investigation of pathway signals between germ cell and granulosa cell.** (**A**) Venn diagram of the common and specific pathway between germ cells and granulosa cells. (**B**) Histogram of the most representative common pathway of germ cells and granulosa cells. (**C** and **D**) Dot plots (**C**) and Vnlplots (**D**) of tight junction related genes in germ cells and granulosa cells. (**E** and **F**) Dot plots (**E**) and Vnlplots (**F**) of adherens junction-related genes in germ cells and granulosa cells. UMAP, uniform manifold approximation projection.
(TIF)

**S7 Fig. Clustering analyses of stromal cell, endothelial cell and immune cell in ovary.** (**A**) Cluster analysis of stromal cells with UMAP plots based on developmental timeline (upper) and cell clusters (below). (**B**) Feature plots of known marker genes of stromal cells. (**C**) Heatmap of top 5 marker genes of stromal cell clusters. (**D**) Cluster analysis of endothelial cells with UMAP plots based on developmental timeline (left) and cell clusters (right). (**E**) Cluster analysis of immune cells with UMAP plots based on developmental timeline (left) and cell clusters (right). The sequencing data was deposited availably in GSE134339, this figure can be produced using scripts at https://github.com/WangLab401/2020scRNA_murine_ovaries. UMAP, uniform manifold approximation projection.
(TIF)

**S1 Table. Marker genes of ovarian cell types during primordial follicle formation.**
(XLSX)

**S2 Table. Top 50 marker genes of germ cell clusters.**
(XLSX)

**S3 Table. Differentially expressed genes of germ cell trajectories.**
(XLSX)

**S4 Table. Enriched GO term of top 100 genes in each gene set for germ cell trajectories.**
GO, Gene Ontology.
(XLSX)

**S5 Table. KEGG pathway of gene sets regard to germ cells in nest and follicle.** KEGG, Kyoto Encyclopedia of Genes and Genomes.
(XLSX)

**S6 Table. Top 50 marker genes of granulosa cell clusters.**
(XLSX)

**S7 Table. Differentially expressed genes of granulosa cells trajectories.**
(XLSX)

**S8 Table. Enriched GO term of top 100 genes in each gene set for granulosa cell trajectories.** GO, Gene Ontology.
(XLSX)

**S9 Table. KEGG pathway of gene sets regard to differentiation of granulosa cells.** KEGG, Kyoto Encyclopedia of Genes and Genomes.
(XLSX)

**S10 Table. KEGG pathways of key genes regard to germ cell and granulosa cell fates.** KEGG, Kyoto Encyclopedia of Genes and Genomes.
(XLSX)

**S1 Data. Original western blot gel images of Fig 3D.**
(TIF)

**S2 Data. The raw data used for quantification of Fig 3D.**
(XLSX)

**S3 Data. The raw data used for quantification of S1B Fig.**
(XLSX)

## Author Contributions

**Conceptualization:** Wei Shen.

**Data curation:** Jun-Jie Wang, Wei Ge, Qiu-Yue Zhai, Xiao-Wen Sun, Wen-Xiang Liu, Lan Li.

**Formal analysis:** Jun-Jie Wang.

**Funding acquisition:** Wei Shen.

**Methodology:** Jun-Jie Wang, Wei Ge.

**Project administration:** Wei Shen.

**Resources:** Wei Shen.

**Supervision:** Wei Shen.

**Validation:** Jun-Jie Wang.

**Visualization:** Jun-Jie Wang.

**Writing – original draft:** Jun-Jie Wang, Chu-Zhao Lei, Paul W. Dyce, Massimo De Felici, Wei Shen.

**Writing – review & editing:** Jun-Jie Wang, Wei Ge, Qiu-Yue Zhai, Jing-Cai Liu, Xiao-Wen Sun, Wen-Xiang Liu, Lan Li, Chu-Zhao Lei, Paul W. Dyce, Massimo De Felici, Wei Shen.

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
