## [Editor Report · Decision Letter 0]

28 Oct 2019

Dear Dr Shen, 

Thank you for submitting your manuscript entitled "Transcriptome Landscape Reveals Underlying Mechanisms of Ovarian Cell Fate Differentiation and Primordial Follicle Assembly" for consideration as a Research Article by PLOS Biology.

Your manuscript has now been evaluated by the PLOS Biology editorial staff as well as by an academic editor with relevant expertise and I am writing to let you know that we would like to send your submission out for external peer review as a METHODS AND RESOURCES paper.

If you agree, before we can send your manuscript to reviewers, we need you to complete your submission by providing the metadata that is required for full assessment. To this end, please login to Editorial Manager where you will find the paper in the 'Submissions Needing Revisions' folder on your homepage. Please click 'Revise Submission' from the Action Links and complete all additional questions in the submission questionnaire.

Please re-submit your manuscript within two working days, i.e. by Oct 30 2019 11:59PM.

Kind regards,

Di Jiang

PLOS Biology

---

## [Decision Letter · Decision Letter 1]

27 Nov 2019

Dear Dr Shen,

Thank you very much for submitting your manuscript "Transcriptome Landscape Reveals Underlying Mechanisms of Ovarian Cell Fate Differentiation and Primordial Follicle Assembly" for consideration as a Methods and Resources at PLOS Biology. I'm handling your manuscript temporarily while my colleague Dr Di Jiang is out of the office. Your manuscript has been evaluated by the PLOS Biology editors, an Academic Editor with relevant expertise, and by three independent reviewers.

The reviews of your manuscript are appended below. You will see that the reviewers find the work potentially interesting. However, based on their specific comments (especially those of reviewer #3, which are relatively critical) and following discussion with the academic editor, I regret that we cannot accept the current version of the manuscript for publication. We remain interested in your study and we would be willing to consider resubmission of a comprehensively revised version that thoroughly addresses all the reviewers' comments. We cannot make any decision about publication until we have seen the revised manuscript and your response to the reviewers' comments. Your revised manuscript would be sent for further evaluation by the reviewers.

Having discussed the reviews with the academic editor, we request that you address all of the points raised by reviewer #3 (as well as the more minor concerns of reviewers #1 and #2). We appreciate that these requests represent a great deal of extra work, and we are willing to relax our standard revision time to allow you six months to revise your manuscript. Please email us (plosbiology@plos.org) to discuss this if you have any questions or concerns, or think that you would need longer than this. At this stage, your manuscript remains formally under active consideration at our journal; please notify us by email if you do not wish to submit a revision and instead wish to pursue publication elsewhere, so that we may end consideration of the manuscript at PLOS Biology.

Your revisions should address the specific points made by each reviewer. Please submit a file detailing your responses to the editorial requests and a point-by-point response to all of the reviewers' comments that indicates the changes you have made to the manuscript. In addition to a clean copy of the manuscript, please upload a 'track-changes' version of your manuscript that specifies the edits made. This should be uploaded as a "Related" file type. You should also cite any additional relevant literature that has been published since the original submission and mention any additional citations in your response. 

Before you revise your manuscript, please review the following PLOS policy and formatting requirements checklist PDF: http://journals.plos.org/plosbiology/s/file?id=9411/plos-biology-formatting-checklist.pdf. It is helpful if you format your revision according to our requirements - should your paper subsequently be accepted, this will save time at the acceptance stage.

Please note that as a condition of publication PLOS' data policy (http://journals.plos.org/plosbiology/s/data-availability) requires that you make available all data used to draw the conclusions arrived at in your manuscript. If you have not already done so, you must include any data used in your manuscript either in appropriate repositories, within the body of the manuscript, or as supporting information (N.B. this includes any numerical values that were used to generate graphs, histograms etc.). For an example see here: http://www.plosbiology.org/article/info%3Adoi%2F10.1371%2Fjournal.pbio.1001908#s5.

For manuscripts submitted on or after 1st July 2019, we require the original, uncropped and minimally adjusted images supporting all blot and gel results reported in an article's figures or Supporting Information files. We will require these files before a manuscript can be accepted so please prepare them now, if you have not already uploaded them. Please carefully read our guidelines for how to prepare and upload this data: https://journals.plos.org/plosbiology/s/figures#loc-blot-and-gel-reporting-requirements.

Upon resubmission, the editors will assess your revision and if the editors and Academic Editor feel that the revised manuscript remains appropriate for the journal, we will send the manuscript for re-review. We aim to consult the same Academic Editor and reviewers for revised manuscripts but may consult others if needed.

If you still intend to submit a revised version of your manuscript, please go to https://www.editorialmanager.com/pbiology/ and log in as an Author. Click the link labelled 'Submissions Needing Revision' where you will find your submission record. 

Sincerely,

Roli Roberts

Roland G Roberts, PhD

Senior Editor

PLOS Biology

on behalf of

Di Jiang, 

Associate Editor

PLOS Biology

REVIEWERS' COMMENTS:

Reviewer #1:

The manuscript by Jun-Jie Wang et al., profiled transcriptomes of fetal ovarian cells during primordial follicle formation by using single-cell RNA sequencing. The manuscript presented a significant amount of data that will serve as a very informative database for understanding many aspects of oocyte differentiation, granulose cell differentiation and primordial follicle assembly. The manuscript is well-written over all and the data are clearly presented. The following issues need be addressed by the authors. 

1. Abstract. Line 24. The timing of primordial follicle assembly is very different in mammals. It takes place perinatally in mice, but around mid-gestation in humans. Please correct this statement. 

2. Introduction. Line 63, “ From here, PGCs move into….between E11.5-E12.5,”. The statement about the timing of PGC migration is not correct. Most PGCs arrive at the gonad by E10.5. (McLaren, 2003, Developmental Biology) 

3. Introduction, Line 66-67. “The oocytes are closely associated in clusters, termed germ cell nests, in which they are connected to each other through cytoplasmic bridges formed…..during mitosis”. There is a fundamental difference between the term “germ cell nest” vs. “germline cyst” (Lei and Spradling, 2013, Development). “Germ cell nest” is used to describe the germ cells that are morphologically clustered. “Germline cyst” refers to sister germ cells that are actually connected by intercellular bridges. The term “oocyte nest” (line 88) does not exist. 

4. Figure 2E, please explain the purpose of this graph? What does the changes in the percentage of cell types mean biologically? 

5. Figure 3F, Why are the percentages of germ cells 25%-based? How was it calculated?

6. Figure 4G-I, How were the candidate genes chosen? Please explain it better in the result section. The same issue was found in Figure 5E, F. 

7. Figure 5B, Figure legends are too small. 

8. Figure 6C. Please explain clusters better, in particular the cluster that is positive for Hes1 and Sox11. Based on the description in the results, Figure 6C is used to understand germ cell development. However, Gata4 and Maf are known transcription factors that are specifically expressed in ovarian somatic cells (De Falco et al., Developmental biology, 2011).

9. There are some minor English issues need to be fixed.

Reviewer #2:

In their manuscript, “Transcriptome Landscape Reveals Underlying Mechanisms of Ovarian Cell Fate Differentiation and Primordial Follicle Assembly,” Wang et al perform single cell RNA-Seq on mouse ovaries across a developmental series (E16.5, PND0, and PND3) to examine the changes in gene expression at the single cell level that accompany oocyte survival and death, granulosa cell differentiation, and crosstalk between oocyte-granulosa cell during primordial follicle assembly. Overall, this study is appears to be well done from a technical standpoint and harnesses the power of innovative new technology to provide further insight into a major developmental transition from germ cell cyst to primordial follicle. However, there are several important ways this study could be improved as outlined below:

1) A better rationale should be provided for why the specific mouse ages were used. For example, how do these ages correlate with the main biological events of GC invasion, cyst breakdown, and primordial follicle assembly.

2) The title overstates the findings of this study. In fact, the authors focus primarily on known pathways. Discovery of known genes and pathways certainly validates this dataset, but does little to advance the field. Although this dataset will be a great resource for the field, a deeper analysis of this dataset should be performed with a focus on novel genes and pathways. The authors focus almost exclusively on the germ and somatic compartments of the ovary and overlook extrafollicular changes. The stromal compartment should be examined, especially given the rise in immune cells beginning at PND 0.

3) Functional validation of novel pathways should be provided. For example, perturbation of pathways involved in key transitions should be performed to demonstrate biological significance. 

4)This manuscript is written for an audience that is familiar with single cell sequencing technology and, therefore, contains a lot of jargon that makes it inaccessible to a general audience. Better explanations should be provided about terminology and methodology – e.g. clusters, pseudotime, regulon activity, SCENIC algorithm. Moreover, more information needs to be provided about how the methods reported here advance what has already been done in the field with respect to single cell sequencing.

5) The authors should discuss the caveats and limitations of this technology and dataset.

6) There are minor grammatical issues throughout the manuscript which should be corrected.

Reviewer #3:

Transcriptome landscape reveals underlying mechanisms of ovarian cell fate differentiation and primordial follicle assembly

Authors Jun-Jie Wang et al. address an unknown facet of oocyte and ovarian somatic cell interactions during germ cell cyst breakdown using single cell sequencing. In order to understand the cell-to-cell interaction in the developing ovary, the authors collected ovaries, in a time-course, at E16.5, P0, and P3 to include germ cell cyst breakdown and the germ cell-to-oocyte transition. In order to understand the role of individual cells during primordial follicle formation, the authors utilized single-cell sequencing to determine the dynamics of germ cells/oocytes and somatic cells/granulosa cells interactions. Included are numerous bioinformatics analyses. The authors determined that during this time course, germ cells were associated with ten distinct clusters and granulosa cells were associated with eight distinct clusters but were best described via pseudo-time and branching points. The interaction between the two cell types were determined by comparing transcriptomes. Overall, the authors corroborate patterns previously reported by numerous studies and have identified pathways that contribute to both germ cell cyst breakdown and the germ cell-to-oocyte transition. The strengths of this manuscript include the utilization of a novel method to extract a large amount of data on germ cells, somatic cells, and their interactions during germ cell cyst breakdown and germ cell-to-oocyte transition. The authors showed that some of their data are supported by other studies and that additional, unexplored pathways may be involved with primordial follicle assembly. Despite of its strengths, there are many major concerns, particularly on the details of the experiments, issues of single cell preparation, and the validity of the cell clusters. These concerns make it difficult to make proper interpretation and conclusion of the results. These concerns are:

1. Figure 1A&B: First, these data have been reported by many others so no new information is provided that warrant the inclusion of such data. Second, information on how these follicle and cyst numbers were obtained was not provided in Materials and Methods or anywhere in the manuscript Third, it is not known where these numbers are consistent with the existing data in the literature. 

2. Figure 1D: How many ovaries were used for each developmental stage? There are more clusters from E16.5 ovary and much less in other two stages. It is possible that more cells at E16.5 increases the granularity for this particular stage. This poses a major issue on the potential bias on clustering and the biological meaning of such clustering. 

3. The information on single cell analysis lacks details. For example, what was the recovery rate of the cells? What were the average barcodes, reads, and genes for each single cell? How many replicates were included? What was the sequencing depth, recovery rate for the cells, and duplets/multiplets? When the single cell sequencing technique started a while ago, it may be acceptable to have only a single sample in the past but now the field has realized that at least a duplicate is necessary. It is unacceptable with just a single sample for each time point. 

4. tSNE used to be the only way for clustering but its reproducibility has become an issue. The field has moved away from tSNE and now uses UMAP for most analyses. I strongly urge the authors to reanalyze all of their data using UMAP. The final results could be quite different from what they have now, particularly in regard to the statistical power that define individual clusters. 

5. Figure 2: There are issues/discrepancy on the clusters/definition of what cell populations they represent and what is already known about the ovary at these stages. Where are the interstitial/stromal cell populations? Interstitial cells represent a significant portion of the cells composing the fetal ovary, and certainly more cells than epithelial cells or erythrocytes. Clusters for interstitial cells should therefore show up at all stages, with an increase in number of cells overtime. For instance, genes such as Nr2f2 (COUPTFII), Acta2 (aSMA), or Tcf21 should be specific / strongly enriched in stromal cell populations. However, these genes were found in granulosa cells (cluster 12), epithelium (clusters 0, 9, 13 for Acta2 and clusters 9 and 12 for Tcf21). This raises the concerns on the single cell preparation and the quality of the cells. 

6. Similarly, the clusters labelled “epithelial cells” represent a high number of cells, up to 30% of all cells at P3 based on Figure 2E. This is not physiologically representative of the population size of epithelial cells, which should correspond to a much smaller cluster. 

7. Figure 2: It is not scientifically sound to use one single sample of single cell clusters to define the percentage of various cell types over time. The case in point is germ cells: how is it possible that the percentage of germ cells at PD0 is more than 50% whereas 20% at E16.5?

8. It is surprising that the % of oocytes / total cells increases from E16.5 to P0, suggesting that the number of oocytes increases during this period, it goes against all previously published quantifications (Lei Lei 2013 ; Grive et al., 2014; Malki et al., 2014 and others). How can the number of oocytes increase when they have already entered meiosis? Could it be a technical problem of dissociation / capture of germ cell populations at E16.5 that skew the % cell distribution? At E16.5, most germ cells are still in nests and physically connected to each other through intercellular bridges. Could it be possible that some elements captured during the single cell experiment at E16.5 and considered as one single germ cell could actually corresponds to several germ cells still connected, and therefore resulting in under-estimating the germ cell % at E16.5?

9. Figure 3: There were so few cells for P3. Even though oocytes go through attrition, there should not be such a huge difference in number of cells when compared to P0. It is possible that so few cells for P3 stage compared to other stages affect the clustering of the germ cells and prevent seeing the heterogeneity present at this stage. 

10. Fig. 3C: Clusters on the heat map could be organized per time-point in order to more easily associate the genes listed to the age of the ovary.

11. Figure 4, 5 and 6: Have the authors confirmed the expression profile of some of the genes using in situ hybridization or immunostaining? All sorts of projections can be generated from single cell data. But it is also glaringly questionable how much biological relevance these projections can actually predict. 

12. Please define “regulon” and “regulon density” in Figure 6. These are not terms known by many readers. 

13. Figure 6A: It is almost impossible (at least to me) to understand what this figure means. The four rows are the top of the figure were never explained. 

14. Fig 6C: purple cluster contains many TF not expressed in the oocytes and are specific of somatic cells. Ex: GATA4, WT1, EMX2. This raises the possible contamination and also multiplets (multiple cells in one droplet). 

15. More explanation and description should be provided on how “granulosa cell” clusters were selected in Figure 7A for further analyses. 

16. The interpretation of the pseudo time analyses for the granulosa cells is questionable. The pre-granulosa cells present at E16.5 (state 1 in Fig.7E) do not give rise to pre-granulosa cells expressing Lgr5/Rspo1. This has been shown in multiple publications. Lgr5+ pre-granulosa cells arise from the ovarian surface epithelium, which is a different cell population than the E16.5 pre-granulosa cells expressing Foxl2 / Cdkn1c. So basically, the pseudo time analyses are performed with cell populations of different sources resulting in bias in the bioinformatic analysis.

17. Figure 8E&G: Has any of these expression pattern been confirmed by in situ hybridization or immunostaining?

18. One of the main purposes of single cell sequencing is to identify novel cell types and pathways. Most of the candidate genes or pathways found in the analysis were previously described. What are the new information/mechanism the derive from this study?

19. Fig S6: the violin plots suggest cross-contaminations between the granulosa populations and the oocyte populations. For instance, Kit is detected in some granulosa cells and Kitl is detected in some oocytes. Same with Gdf9, detected in some granulosa. These data suggest a problem with the single cell data, probably at the cell dissociation step. This likely impacted the clustering and data analyses.

20. This manuscript lacks mention/definition of atretic germ cells or somatic cells during the time course. How were these cell types taken into consideration into the analyses? Were the atretic cells omitted entirely? Please address in the discussion.

Editorial comments:

Line 50: The ovarian reserve is not established yet at birth, but rather about a week after birth. At birth, formation of all primordial follicles is not complete and some oocytes are being lost through attrition.

Line 141: you need to explain what canonical correlation analysis is so that the reader understands the meaning of these results. There is not sufficient information on Figures S1D and S1E to understand what they represent.

Line 161: the terms “late-follicular” / “Late follicle” are inappropriate and could be confusing since the latest stage analyzed here, P3, still represents very early stages of folliculogenesis. The authors should use a different terminology for the stages of oocyte development they describe in figure 3/4. Maybe “Pre follicle / early follicle formation / late follicle formation”.

Fig.4E, 5B-E-F, 6A, 7D-E-F-H: the axis or legend of the graphs are not visible/too small.

Line 195: “Conversely, ClueGO functional networks of genes in cluster 3 and 4 under regulated in germ cell cysts…” Cluster 3 represents genes highly expressed in germ cell cysts as mentioned line 183. Clusters 3 and 4 have completely different profiles, merging them for gene network analysis does not make sense.

Line 246: typo Cdkn1c.

The single branch point in the pseudo-time analysis of the granulosa cells was not well-defined. Please define the potential biological significance of the single branch point.

Numerous bioinformatic approaches were taken to analyze the data, but many graphs are not explained nor contribute to significant discussion points such as Figure 4E, 5B, 6C, and 6D. Replace these figures to the supplementary or discuss the significance of these graphs’ data.

The final figure, Figure 9, is not clear. The legend on the bottom indicating expression levels between high and low in orange and green are not clearly represented in the figure. The study’s time course should be included (E16.5, P0, P3).

Within the manuscript, vague terminology such as “they” and “these” are used and should not be utilized in scientific writing. Remove and replace the use of these words. Lines 126, 284, and 361.

A conclusion paragraph is lacking and must be included in the discussion. Specifically address the overall impact of the research, how these data compare to other scRNAseq studies (such as Nef), the overall strengths and weaknesses of this paper, and what else needs to be done in order to address the molecular control underlying primordial follicle assembly.

In figures 2D and 3E, please including a legend defining the size of the dots in the dot plot.

Line 226, it is not clear what the authors mean “…despite their expression being limited to specific regions.” Please elaborate as Figure 6C appears to show that the four transcription factors are throughout the heatmap.

Line 383, please elaborate “…gene expression changes are not always essential.” as this manuscript focuses on RNA sequencing data.

Lines 427 – 432 is an unnecessarily long sentence that needs to be broken up.

Figure S2C, the scale values are not clear and overlap each other.

Figure 1A, provide scale bar for zoomed in IF images on the far right.

Figure S5C, the scale value is not defined and appears to be missing a value.

Line 841, do the authors mean “Venn diagram” and not “Vlnplots”?

The legend in figures 4E and 5B are illegible. Increase the size of the figure legend.

---

## [Decision Letter · Decision Letter 2]

10 Nov 2020

Dear Dr Shen,

Thank you for submitting your revised Methods and Resources entitled "Single-cell transcriptome landscape of ovarian cells during primordial follicle assembly in mice" for publication in PLOS Biology. I have now obtained advice from two of the original reviewers and have discussed their comments with the Academic Editor. 

Based on the reviews, we will probably accept this manuscript for publication, assuming that you will modify the manuscript to address the remaining points raised by the reviewers. Please also make sure to address the data and other policy-related requests noted at the end of this email.

IMPORTANT:

a) Please attend to my Data Policy and Ethics Policy requests further down the email.

b) Please could you add one or two sentences to your Abstract, stating what novel biological insights are provided by this study (or might be provided by further studies that use this data)?

c) Please address the remaining concerns raised by the two reviewers.

We expect to receive your revised manuscript within two weeks. Your revisions should address the specific points made by each reviewer. In addition to the remaining revisions and before we will be able to formally accept your manuscript and consider it "in press", we also need to ensure that your article conforms to our guidelines. A member of our team will be in touch shortly with a set of requests. As we can't proceed until these requirements are met, your swift response will help prevent delays to publication.

- a cover letter that should detail your responses to any editorial requests, if applicable

*Copyediting*

*Published Peer Review History*

*Early Version*

Sincerely,

Roli Roberts

Senior Editor,

rroberts@plos.org,

PLOS Biology

ETHICS STATEMENT:

Your ethics statement says "“Animals were housed according to the national guideline and Ethical Committee of Qingdao Agricultural University.” My understanding is that all that you did was to mate the mice and then sacrifice them, which would need no further approvals, but please could you specify how the mice were sacrificed?

DATA POLICY:

Regardless of the method selected, please ensure that you provide the individual numerical values that underlie the summary data displayed in the following figure panels as they are essential for readers to assess your analysis and to reproduce it: Figs 1BCDE, 2ABCD, 3D, 4ABCDEG, 5ABCDE, 6ABCDEFG, 7ABCDEFGH, S1BDE, S2ABCDEFG, S3ABCDE, S4ABCD, S5ABCD, S6BCDEFG, S7ABCDE. NOTE: the numerical data provided should include all replicates AND the way in which the plotted mean and errors were derived (it should not present only the mean/average values).

Reviewer remarks:

REVIEWERS' COMMENTS:

Reviewer #2:

The authors have addressed my comments satisfactorily and I believe the revised manuscript is much strengthened. I just have one minor request which is that they authors integrate the complete rationale for the three ages used for collecting cells in the actual manuscript text. In the reviewer response, the authors provided an important explanation about the relevant biology which is lacking in the actual manuscript text. 

Reviewer #3:

In the revised manuscript, the authors took into account our comments and completely reanalyzed their single-cell datasets, resulting in a significantly changed manuscript. 

A major concern about the previous version was that the scRNA-seq data clustering of the cell populations was not reflecting the biology of the ovary. Particularly, some clusters appeared to be composed of a mixture of different known cell populations, which skewed all the downstream analyses of these clusters. It resulted in some conclusions that were not supported by what is already known in the field. In this new version of the manuscript, the authors used more stringent settings/new methods to reanalyze their scRNA-seq data. It greatly improved the identification of the cell populations and downstream analyzes of each population. Consequently, all the figures of the manuscript are completely changed. The authors also took in account the reviewers comment of focusing on novelty rather than what is already known. A validation for new markers for identified sub-population of cells is now provided (new Figure 3). 

The new combined analysis of germ cell and granulosa for identification of cross-talks of ligand/receptor of common signaling pathways in Fig. S5 is interesting, although it could have been used to characterize new and less known pathways in addition to the already well characterized pathways presented here. Some validation of these findings through immuno-fluorescence or ISH would enhance the confidence of the scRNA-seq findings. The authors also provide more detailed explanations and definitions throughout the manuscript so that non-expert readers can better understand some sc-RNA-seq jargon used. Overall, the authors have greatly improved the manuscript, for the analysis of the data, their interpretation and explanations of the results.

My specific comments are:

1. Line 29: "the complete dynamic genetic programs of germ and granulosa cells from E16.5 to PD3 were reported for the first time, to our knowledge": This is an overstatement as a recent publication by Niu and Spradling have provided a time course of sc-RNA-seq in ovaries at E11.5, E12.5, E14.5, E16.5, E18.5 P1, P5 and also included specific analyzes of both germ cells and granulosa cells (Reference 47 in the manuscript). Since the current manuscript and this published paper both focus on very similar subjects, it would be relevant to have a few sentences in the discussion that specifically compares findings between the 2 manuscripts.

2. Line 135: "Furthermore, the cell cycle phase score result showed the germ cells heterogeneity was not interfered by cell cycle genes expression (S2A Fig)": At E16.5-P3 the germ cells have all initiated meiosis. Why the analysis based on cell cycle when germ cell are in meiosis?

3. I find the pseudotime analyses misleading, particularly on the interpretation of what the sub-populations of both germ cells and granulosa cell actually are. For germ cells, the authors stated in Line 160 "In such pseudotime trajectories, three branches implied three differentiated fates of germ cells". This definition does not apply to these sub-populations of germ cells. There is continuous differentiation into these sub-populations, from State 1 to State 3 (although it is unclear what State 2 actually represents). State 1 and State 3 are not different "fates of differentiation", but rather different "stages of differentiation" within the same population. State 1 is defined as "Nest" germ cells and state 3 is defined as "Follicle" germ cells. What does State 2 represent? Is it an intermediary state between state 1 and 3, or is it possible it could represent some pre-atretic germ cells? Gene set 3 seems to be specific of this State 2 and includes GO terms related to "cellular response to DNA damage" (Fig. 4D-E).

4. Following comment 3 for granulosa cells, the State 1, which were defined as "Early PGs", are actually BPGs, but at a less differentiated state than State 3 labelled "BPGs". Both represent wave 1 of granulosa cells. Maybe it would be more appropriate to name State 1 "Early BPGs" and State 3 "Differentiating BPGs". State 2 "EPGs" is defined line 225 as "differentiated pre-granulosa cells". On the contrary, these cells represent wave 2 of granulosa cells that are at an even earlier stage of differentiation: these cells are currently recruited from the epithelium and do not express Foxl2 yet in contrary to States1-3, but they eventually will to form the cortical follicles.

5. Fig. 5A-E: what do the numbers in parentheses next to the regulons mean? Ex: "Foxo3 (16g)". It is not defined in the legend.

6. Line 214: "For example, Wnt4 expression was most activated at E16.5, which may imply the differentiation of ovarian pre-granulosa cell and preparation the secondary follicle formation [19,40]." I don't understand this sentence. What is the link to secondary follicle formation?

7. Fig. S6C. I don't understand the point of the combined clustering of granulosa and germ cells when there is already the transcriptomic data for each separated population. 

8. Fig. 7I: FOXO3 is expressed in the nuclei of inactive oocytes. It will only be translocated in the cytosol when oocytes are activated when primary follicles form. So most oocytes should have nuclear FOXO3. At P3, majority of the oocytes would have FOXO3 located in the nucleus and a few oocytes in the medulla should have FOXO3 located in the cytosol. Similarly, TAZ is a transcription factor. It is supposed be located in the nucleus.

9. Fig. S7 about analyses of other somatic cells is not mentioned at all in the results. This figure should be described in the result part. The rational of the authors to not focus on interstitium populations is: "However, due to the limited information available regarding other ovarian somatic compartments, the current study placed most attention on germ cells and granulosa cells." On the contrary, this actually would make a good argument to take advantage of these new single cell data to further study these cells populations that are less known than the germ cells and granulosa. Since the authors provide this figure S7, they should talk about it. This would bring more novelty to the study.

10. Most parts of the discussion are repetitions of results and including detailed lists of genes that should be only described in the result part. The discussion should be shorten and focus on the take-home messages and bigger picture of these results in the context of what is known in the field and what novelty this study provides.

---

## [Editor Report · Decision Letter 3]

3 Dec 2020

Dear Dr Shen,

On behalf of my colleagues and the Academic Editor, Polina V Lishko, I am pleased to inform you that we will be delighted to publish your Methods and Resources in PLOS Biology. 

PRODUCTION PROCESS

Before publication you will see the copyedited word document (within 5 business days) and a PDF proof shortly after that. The copyeditor will be in touch shortly before sending you the copyedited Word document. We will make some revisions at copyediting stage to conform to our general style, and for clarification. When you receive this version you should check and revise it very carefully, including figures, tables, references, and supporting information, because corrections at the next stage (proofs) will be strictly limited to (1) errors in author names or affiliations, (2) errors of scientific fact that would cause misunderstandings to readers, and (3) printer's (introduced) errors. Please return the copyedited file within 2 business days in order to ensure timely delivery of the PDF proof. 

If you are likely to be away when either this document or the proof is sent, please ensure we have contact information of a second person, as we will need you to respond quickly at each point. Given the disruptions resulting from the ongoing COVID-19 pandemic, there may be delays in the production process. We apologise in advance for any inconvenience caused and will do our best to minimize impact as far as possible.

EARLY VERSION

PRESS 

Kind regards,

Erin O'Loughlin

Publishing Editor 

PLOS Biology

on behalf of

Roland Roberts,

Senior Editor

PLOS Biology